# Histone H1 prevents non-CG methylation-mediated small RNA biogenesis in *Arabidopsis* heterochromatin

**Jaemyung Choi[1†], David B Lyons[1†], Daniel Zilberman[1,2]***

[1]Department of Cell and Developmental Biology, John Innes Centre, Norwich, United Kingdom; [2]Institute of Science and Technology, Klosterneuburg, Austria

**Abstract** Flowering plants utilize small RNA (sRNA) molecules to guide DNA methyltransferases to genomic sequences. This RNA-directed DNA methylation (RdDM) pathway preferentially targets euchromatic transposable elements. However, RdDM is thought to be recruited by methylation of histone H3 at lysine 9 (H3K9me), a hallmark of heterochromatin. How RdDM is targeted to euchromatin despite an affinity for H3K9me is unclear. Here, we show that loss of histone H1 enhances heterochromatic RdDM, preferentially at nucleosome linker DNA. Surprisingly, this does not require SHH1, the RdDM component that binds H3K9me. Furthermore, H3K9me is dispensable for RdDM, as is CG DNA methylation. Instead, we find that non-CG methylation is specifically associated with sRNA biogenesis, and without H1 sRNA production quantitatively expands to non-CG-methylated loci. Our results demonstrate that H1 enforces the separation of euchromatic and heterochromatic DNA methylation pathways by excluding the sRNA-generating branch of RdDM from non-CG-methylated heterochromatin.

***For correspondence:**
daniel.zilberman@ist.ac.at

[†]These authors contributed equally to this work

## Introduction

Transposable elements (TEs) and their remnants comprise a substantial fraction of eukaryotic genomes and generally must be kept silent to ensure genome integrity and function (*Bourque et al., 2018*). TE silencing is achieved despite the disruption caused by each cell division, whereby half of the genome and histone proteins are made anew. Robust cellular memory of the inactive state is achieved by feedback interactions that reinforce and concentrate chromatin features and factors that contribute to transcriptional silencing and exclude activating factors (*Allshire and Madhani, 2018*; *Zhang et al., 2018b*). However, silent chromatin domains are not homogenous. Flowering plants have two major types of TE-associated silent chromatin: GC-rich coding regions of autonomous TEs, and AT-rich chromatin comprised of gene-proximal TE remnants, short nonautonomous TEs, and edges of autonomous TEs (*Sequeira-Mendes et al., 2014*; *To et al., 2020*; *Zemach et al., 2013*; *Zhong et al., 2012*). Although both are comprised of TEs, these chromatin types have distinct features (*Sequeira-Mendes et al., 2014*; *Zemach et al., 2013*). How two types of silent TE chromatin are distinguished and kept separate within the nucleus is a major open question.

Both types of TE chromatin feature extensive cytosine methylation in the CG context catalyzed by MET1 (plant homolog of Dnmt1) (*Cokus et al., 2008*; *Lister et al., 2008*; *Zemach et al., 2013*), and are also methylated at non-CG (CHG and CHH, where H is A, T, or C) cytosines (*Stroud et al., 2014*; *Zemach et al., 2013*). GC-rich TE sequences have high levels of histone modifications associated with heterochromatin, including methylation of lysine nine of histone H3 (H3K9me), and are therefore known as heterochromatic TEs (*Sequeira-Mendes et al., 2014*; *Zemach et al., 2013*). Non-CG methylation (mCH) at heterochromatic TEs is catalyzed primarily by chromomethylases (CMTs; CMT3 for CHG methylation and CMT2 for CHH), which are recruited to H3K9 dimethylated (H3K9me2) nucleosomes by histone-tail-interacting domains (*Du et al., 2012*; *Stoddard et al., 2019*; *Stroud et al.,*

**eLife digest** Cells adapt to different roles by turning different groups of genes on and off. One way cells control which genes are on or off is by creating regions of active and inactive DNA, which are created and maintained by different groups of proteins. Genes in active DNA regions can be turned on, while genes in inactive regions are switched off or silenced. Silenced DNA regions also turn off 'transposable elements': pieces of DNA that can copy themselves and move to other regions of the genome if they become active. Transposons can be dangerous if they are activated, because they can disrupt genes or regulatory sequences when they move.

There are different types of active and inactive DNA, but it is not always clear why these differences exist, or how they are maintained over time. In plants, such as the commonly-studied weed *Arabidopsis thaliana*, there are two types of inactive DNA, called E and H, that can silence transposons. In both types, DNA has small chemicals called methyl groups attached to it, which help inactivate the DNA. Type E DNA is methylated by a process called RNA-directed DNA methylation (RdDM), but RdDM is rarely seen in type H DNA.

Choi, Lyons and Zilberman showed that RdDM is attracted to E and H regions by previously existing methylated DNA. However, in the H regions, a protein called histone H1 blocks RdDM from attaching methyl groups. This helps focus RdDM onto E regions where it is most needed, because E regions contain the types of transposons RdDM is best suited to silence.

When Choi, Lyons and Zilberman examined genetically modified *A. thaliana* plants that do not produce histone H1, they found that RdDM happened in both E and H regions. There are many more H regions than E regions, so stretching RdDM across both made it less effective at silencing DNA.

This work shows how different DNA silencing processes are focused onto specific genetic regions, helping explain why there are different types of active and inactive DNA within cells. RdDM has been studied as a way to affect crop growth and yield by altering DNA methylation. These results may help such studies by explaining how RdDM is naturally targeted.

2014; *Zemach et al., 2013*). SUVH family H3K9 methyltransferases are in turn recruited to methylated DNA via SRA domains, forming a self-reinforcing loop (*Du et al., 2014*; *Johnson et al., 2007*; *Rajakumara et al., 2011*). *Arabidopsis thaliana* plants lacking functional chromomethylases (*cmt2cmt3* mutants) almost completely lack mCH at heterochromatic TEs, and their H3K9 methylation is greatly reduced (*Stroud et al., 2014*).

AT-rich TE sequences are low in H3K9me and other heterochromatic histone modifications, and are therefore known as euchromatic TEs (*Sequeira-Mendes et al., 2014*; *Zemach et al., 2013*). In contrast to the SUVH/CMT feedback loop that predominates in heterochromatin, RNA-directed DNA methylation (RdDM) catalyzes cytosine methylation within euchromatic TEs (*Zemach et al., 2013*; *Zhong et al., 2012*). RdDM loci are transcribed by a methylation-tolerant RNA polymerase II derivative (Pol IV) that couples cotranscriptionally with RNA-dependent RNA polymerase 2 (RDR2) to make double stranded RNA, which is processed into 23/24-nt fragments by Dicer-like 3 (DCL3) (*Singh and Pikaard, 2019*). These 24-nt small RNAs (sRNA) are subsequently denatured and loaded into Argonaute (AGO) protein complexes. AGO–sRNA complexes associate with another Pol II family enzyme, Pol V, to recruit Domains Rearranged Methylases (DRMs; primarily DRM2 in *Arabidopsis*) (*Erdmann and Picard, 2020*; *Matzke and Mosher, 2014*; *Raju et al., 2019*; *Wendte and Pikaard, 2017*).

Like the SUVH/CMT pathway, RdDM comprises positive feedback loops. Pol V is recruited to methylated DNA, effectively seeking its own product (*Liu et al., 2014*; *Wongpalee et al., 2019*; *Zhong et al., 2012*). A more paradoxical feedback loop is thought to involve recruitment of Pol IV to H3K9me (*Erdmann and Picard, 2020*; *Matzke and Mosher, 2014*; *Raju et al., 2019*; *Wendte and Pikaard, 2017*). This hypothesis emerged from the observation that Pol IV-mediated sRNA production at many loci requires SHH1/DTF1, a protein that binds H3K9me2 and monomethylated H3K9me (H3K9me1) in vitro (*Law et al., 2013*; *Zhang et al., 2013*). This model of Pol IV recruitment necessitates explaining how RdDM in general, and Pol IV specifically, is excluded from heterochromatic TEs with high H3K9me and targeted to euchromatic TEs with low H3K9me. Reliance of Pol IV on H3K9me also poses two theoretical questions. First, why would RdDM depend on a core component of the SUVH/CMT feedback loop (H3K9me2), when the two DNA methylation systems have largely nonoverlapping primary

targets (*Stroud et al., 2014*), and RdDM targets are H3K9me depleted? Second, the euchromatic TEs targeted by RdDM are often comprised of just one or two nucleosomes (*Zemach et al., 2013*). Maintenance of histone modifications is expected to be unstable at such short sequences due to the random partitioning of nucleosomes to sister chromatids following DNA replication (*Angel et al., 2011*; *Berry and Dean, 2015*; *Lövkvist and Howard, 2021*; *Ramachandran and Henikoff, 2015*; *Zilberman and Henikoff, 2004*). Why would RdDM, a pathway capable of almost nucleotide-level resolution (*Blevins et al., 2015*; *Zhai et al., 2015*) and specialized for silencing short TEs, be tied to a histone modification that requires longer sequences for stable propagation?

Here, we show that Pol IV activity is recruited to sequences with non-CG DNA methylation regardless of H3K9me, so that both the Pol IV and Pol V branches form positive feedback loops with the ultimate product of RdDM. We also show that linker histone H1 impedes RdDM activity in GC-rich heterochromatin, thereby restricting RdDM to AT-rich euchromatic TEs. We propose that without H1, RdDM would be diluted into and effectively incapacitated by the vast stretches of non-CG-methylated heterochromatin common in plant genomes (*Feng et al., 2010*; *Niederhuth et al., 2016*; *Ritter and Niederhuth, 2021*; *Zemach et al., 2010*). The affinity of H1 for GC-rich heterochromatin (*Choi et al., 2020*) focuses RdDM activity on short, AT-rich euchromatic TEs that RdDM is uniquely suited to silence.

## Results

### Histone H1 levels predict the global bifurcation of mCH pathways

To understand how the CMT and RdDM pathways are separated, we categorized *Arabidopsis* TEs by the dependence of their CHH methylation (mCHH) either on CMT2 (CMT TEs) or DRM2 (DRM TEs). Among 18784 TEs with more than 2% mCHH in wild-type (*wt*) plants, 4486 TEs were demethylated in *cmt2* plants and 3039 TEs lost mCHH in *drm2* (mCHH in the mutants <0.02, Fisher's exact test $p <$ 0.01, TEs longer than 200 bp; *Figure 1—figure supplement 1A* and *Figure 1—source data 1*). Only 80 TEs had mCHH diminished below 2% in both mutants (*Figure 1—source data 1*), consistent with the largely separate sets of primary DRM and CMT targets (*Sigman and Slotkin, 2016*; *Stroud et al., 2014*).

Next, we used random forest classification (*Breiman, 2001*; *Ishwaran et al., 2012*) to identify predictors of DRM or CMT targets (*Figure 1A*). We included genetic and epigenetic features known to be associated with RdDM or CMT activity, as well as linker histone H1. H1 is specifically enriched in heterochromatic TEs, and its loss leads to increased DNA methylation at heterochromatic TEs and decreased methylation at euchromatic ones (*Bourguet et al., 2021*; *Lyons and Zilberman, 2017*; *Papareddy et al., 2020*; *Rutowicz et al., 2015*; *Zemach et al., 2013*). As expected, sRNA abundance can distinguish CMT and DRM TEs (*Figure 1A*). H3K9me1 is also a good classifier (*Figure 1A*). However, the best classifier turned out to be H1 (*Figure 1A*). Using all variables in *Figure 1A*, we could predict CMT and DRM TEs with an error rate of 2.15% (*Figure 1B*). With just H3K9me1 and H1, the prediction is almost as accurate (5.42% error; *Figure 1B*). Remarkably, H1 alone successfully identifies CMT and DRM TEs (12.17% error; *Figure 1B*), suggesting that H1 is fundamental to separating these silencing pathways.

### RdDM activity relocates to heterochromatin without H1

To understand how H1 regulates the CMT and DRM pathways, we analyzed 24-nt sRNA expression, DNA methylation, and H3K9me2 in *h1* plants that have inactivating mutations in both of the canonical *Arabidopsis* H1 genes (*Zemach et al., 2013*). Consistent with published results (*Bourguet et al., 2021*; *Lyons and Zilberman, 2017*; *Papareddy et al., 2020*; *Rutowicz et al., 2015*; *Zemach et al., 2013*), we found an elevation of CHG methylation (mCHG), H3K9me2 and mCHH at CMT TEs (*Figure 1C, D*). CMT TEs are depleted of sRNAs in *wt* leaves, but sRNA expression increases 5.6-fold in *h1* plants (*Figure 1D*, *Figure 1—figure supplement 1B, C*). sRNA expression in *h1* positively correlates with that in *wt* (*Figure 1—figure supplement 1B*), indicating that loss of H1 amplifies sRNA expression at RdDM-capable CMT TEs rather than creating de novo RdDM targets.

In contrast to the hypermethylation of CMT TEs, DRM TEs lose H3K9me2, mCHG, mCHH, and sRNA expression in *h1* plants (*Figure 1C, D* and *Figure 1—figure supplement 1D*). Despite the loss of sRNA at DRM TEs, global 24-nt sRNA abundance is not altered in *h1* plants (*Figure 1—figure*

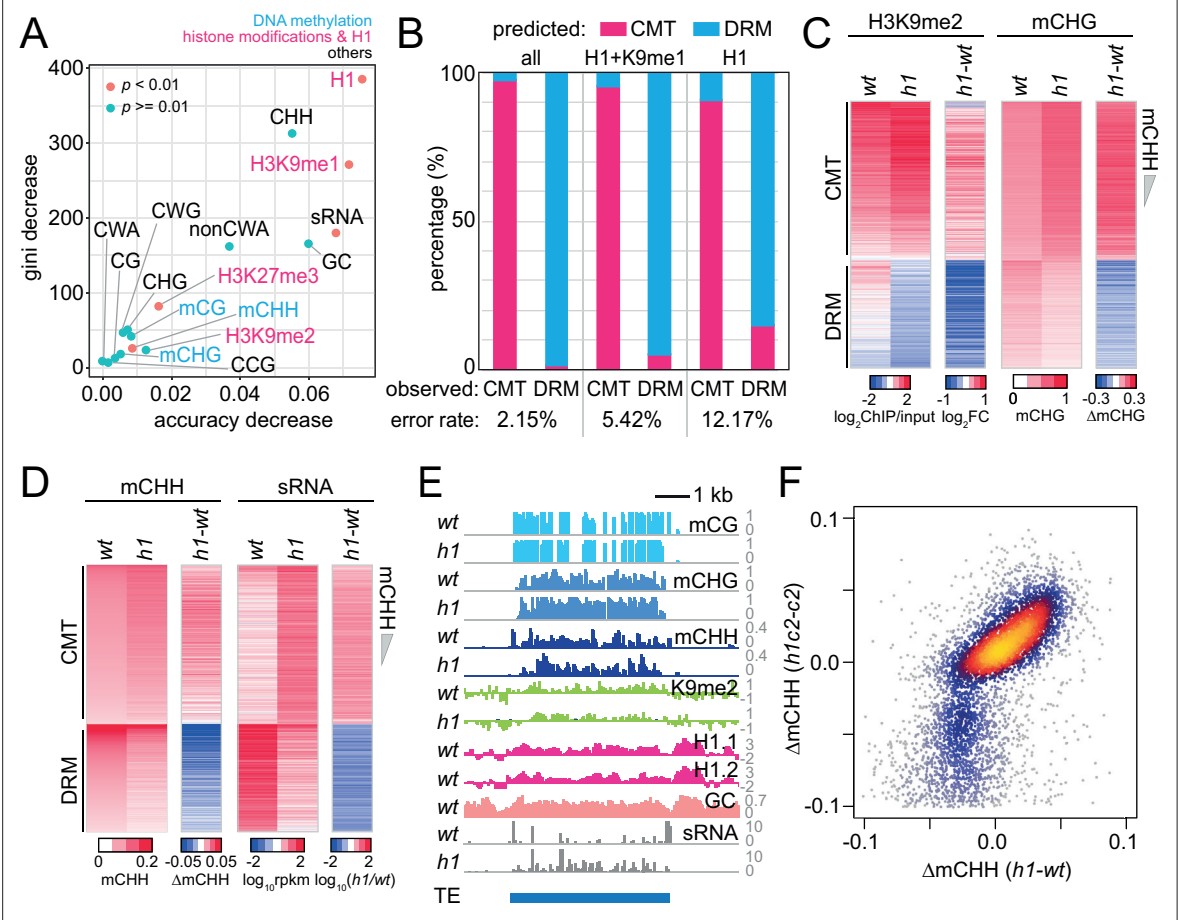

**Figure 1.** Histone H1 prohibits RNA-directed DNA methylation of chromomethylase (CMT)-dependent heterochromatic transposons. (**A**) The importance of DNA methylation, histone H3 modifications, small RNA (sRNA), H1, and cytosine sequence context to predict CMT transposable element (TE) or DRM TE classes by random forest classification. (**B**) Prediction of CMT or DRM TE classes by random forest classification with all variables, H1 and H3K9me1, or only H1. Heatmaps of H3K9me2 and CHG methylation (mCHG) levels (**C**) and mCHH and sRNA levels (**D**) at CMT and DRM TEs in *wt* and *h1* plants. TEs were sorted by mCHH level in *wt*. (**E**) Example of DNA methylation and sRNA expression at a CMT TE in *wt* and *h1* (AT1TE58075). (**F**) mCHH difference between *wt* and *h1* (*x*-axis) vs *h1cmt2* (*h1c2*) and *cmt2* (*c2*; *y*-axis) at CMT TEs.

The online version of this article includes the following figure supplement(s) for figure 1:

**Source data 1.** Lists of chromomethylase (CMT)- and DRM-dependent transposons and intermediate transposons in *Arabidopsis*.

**Figure supplement 1.** Chromomethylase (CMT) transposable elements (TEs) gain non-CG DNA methylation and small RNA (sRNA) expression in *h1* plants.

---

*supplement 1E*), indicating the reallocation of RdDM activity from DRM to CMT TEs. This phenomenon can be observed within individual TEs, with sRNA biogenesis and mCHH relocating from the AT-rich edges in *wt* to the GC-rich internal sequences in *h1* (*Figure 1E*). The relocation of sRNA production and mCHH into TE interiors in *h1* plants is also apparent in aggregate at TEs that retain substantial mCHH in *drm2* and *cmt2* mutants (intermediate TEs that are not classed either as DRM or CMT TEs; *Figure 1—figure supplement 1A and F, G*). CMT TE mCHH increases to the same relative extent in *h1* plants devoid of CMT2 (*h1c2*; *Figure 1F* and *Figure 1—figure supplement 1H*), indicating that mCHH hypermethylation at CMT TEs in *h1* mutants is caused by RdDM. These results indicate that RdDM relocates into heterochromatin in the absence of H1 and are consistent with recently published work (*Bourguet et al., 2021*; *Papareddy et al., 2020*).

## Lack of H1 promotes sRNA biogenesis in linker DNA

Absence of H1 in *Arabidopsis* causes a preferential increase of heterochromatic TE DNA methylation within linker DNA, the regions between nucleosomes (*Lyons and Zilberman, 2017*). The average

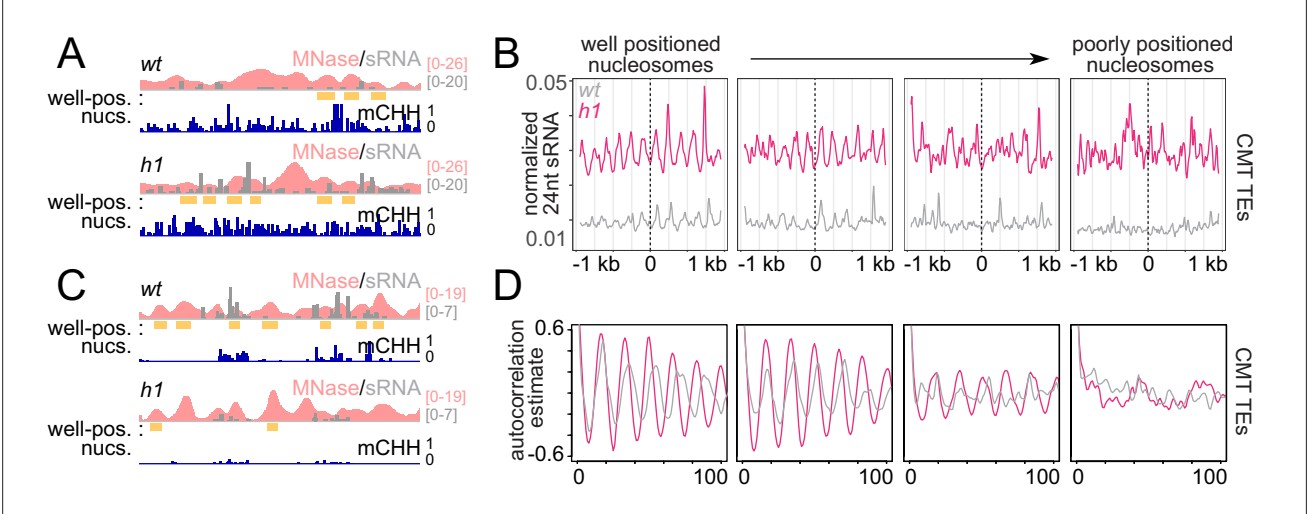

**Figure 2.** RdDM is preferentially active in linker DNA in *h1* plants. (**A, C**) Example of a chromomethylase (CMT) transposable element (TE; Chr2: 3,944,600–3,946,400) (**A**) and DRM TE (Chr2: 6,389,500–6,392,500) (**C**) with well-positioned nucleosomes (yellow boxes). Smoothed MNase-seq (apricot), sRNA expression (gray), and mCHH (indigo) are plotted. (**B**) Average sRNA expression around well positioned or poorly positioned nucleosomes at CMT TEs. (**D**) Autocorrelation estimates of average sRNA values shown in (**B**) to illustrate shortened small RNA (sRNA) phasing in *h1* mutants corresponding to shortened nucleosome repeat length. Nucleosome positioning data and designations are from *Lyons and Zilberman, 2017*.

The online version of this article includes the following figure supplement(s) for figure 2:

**Figure supplement 1.** RNA-directed DNA methylation (RdDM) is enriched at linker DNA around well-positioned nucleosomes.

distance between heterochromatic nucleosomes is also reduced from ~180 to 167 bp (*Choi et al., 2020*). Given the relative promiscuity of RNA Pol IV initiation (*Zhai et al., 2015*) and the increased sRNA abundance at CMT TEs in *h1* (*Figure 1D*, *Figure 1—figure supplement 1B, C*), we asked whether patterns of sRNA production with respect to nucleosomes are altered in *h1*. As expected, overall levels of sRNA are increased around nucleosomes of CMT TEs and decreased at DRM TEs (*Figure 2A–C* and *Figure 2—figure supplement 1*). An overt sRNA linker bias is apparent in both *h1* and *wt* around the best-positioned nucleosomes (*Figure 2A–C* and *Figure 2—figure supplement 1*). This pattern becomes less obvious at less-well-positioned loci until it disappears completely (*Figure 2B* and *Figure 2—figure supplement 1*), as illustrated by measuring the correlation of the sRNA signal to itself (*Figure 2D*). The shortening *h1* sRNA autocorrelation around better positioned nucleosomes (*Figure 2D*) demonstrates how the linker histone dictates sites of sRNA production directly through linker occlusion and indirectly through nucleosome positioning.

## sRNA biogenesis is associated with H3K9me and mCH

Because H3K9me is thought to recruit Pol IV activity (*Law et al., 2013*; *Zhang et al., 2013*), we investigated how sRNA distribution changes in relation to H3K9me1/2 in *h1* plants. In *wt*, sRNA expression increases as H3K9me1 and H3K9me2 levels rise, but this trend reverses at TEs with more H3K9me and H1 (*Figure 3A, B*). In contrast, sRNA expression shows a relatively simple, direct relationship with H3K9me1 and H3K9me2 in *h1* plants (*Figure 3A, B*), suggesting that H1 prevents Pol IV from following the H3K9me gradient.

Unlike TEs, gene bodies normally have low levels of H3K9me, mCH, and sRNA (*Figure 3C*; *Zhang et al., 2018b*). However, many genes gain H3K9me and mCH (especially mCHG) in plants lacking the H3K9 demethylase IBM1 (*Miura et al., 2009*). Although this hypermethylation does not require RdDM (*Inagaki et al., 2010*; *Saze et al., 2008*), recruitment of Pol IV by H3K9me would predict sRNA biogenesis in *ibm1* genes. Indeed, we find increased sRNA and mCHH levels in *ibm1* genes associated with high H3K9me2 and mCHG (*Figure 3C, D*). Hence, the presence of H3K9me or mCH may be sufficient to trigger 24-nt sRNA production.

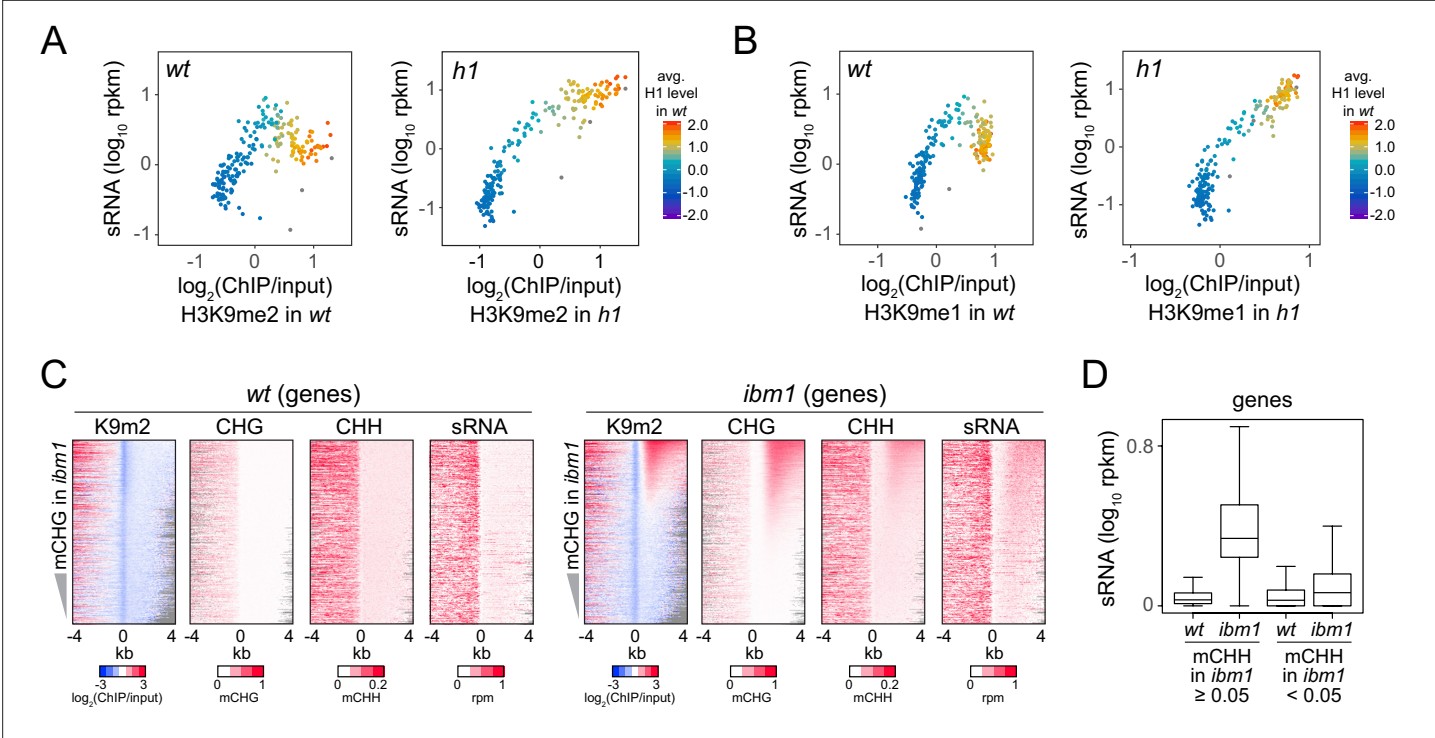

**Figure 3.** Small RNA (sRNA) biogenesis is associated with H3K9me and non-CG methylation. (**A**, **B**) Average H3K9me2 (**A**) or H3K9me1 (**B**) (x-axis) and sRNA expression level (y-axis) in *wt* and *h1*. Each dot represents the average of 100 transposable elements (TEs) sorted by GC content. (**C**) Distribution of H3K9me2, non-CG methylation, and sRNA expression around 5′ ends of genes in *wt* and *ibm1* plants. (**D**) A boxplot shows sRNA expression level at genes in *wt* and *ibm1* plants. Genes that have more than 5% mCHH in *ibm1* or less than 5% mCHH in *ibm1* are plotted separately.

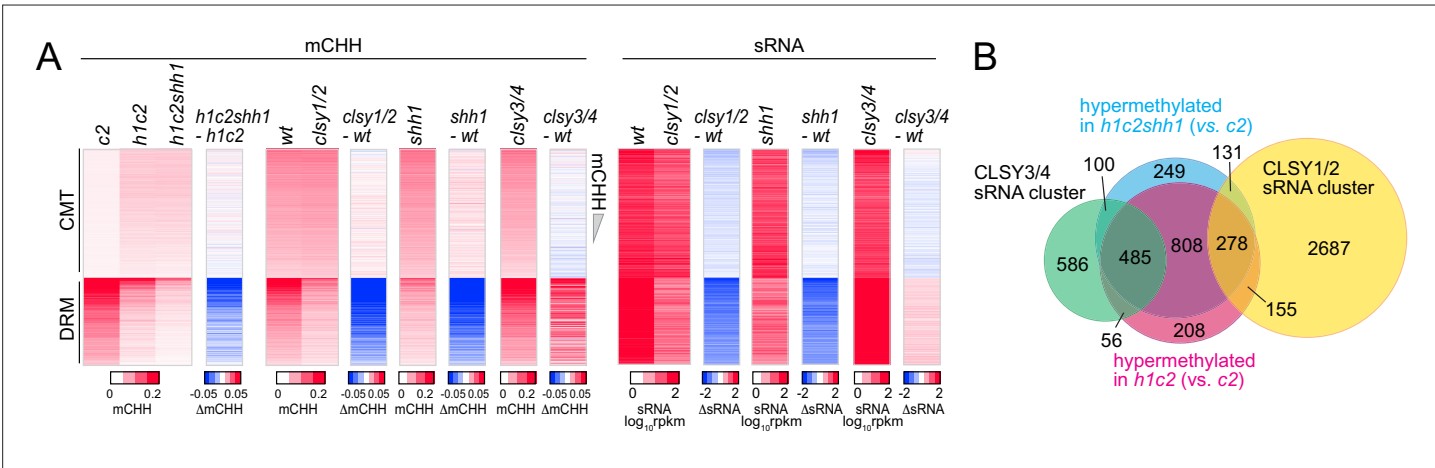

**Figure 4.** SHH1 is not required for non-CG hypermethylation in *h1*. (**A**) Heatmaps of mCHH and small RNA (sRNA) expression at chromomethylase (CMT) and DRM transposable elements (TEs) in plants with *shh1* or *clsy* mutations. (**B**) Venn diagram of TEs in indicated categories.

The online version of this article includes the following figure supplement(s) for figure 4:

**Figure supplement 1.** Loss of *h1* causes chromomethylase (CMT)2-independent hypermethylation at transposable elements (TEs) with CLSY3/4 small RNA (sRNA) clusters.

**Figure supplement 2.** Heatmaps of *wt* sRNA expression at chromomethylase (CMT), DRM transposable elements (TEs) and CLSY1/2, CLSY3/4 small RNA (sRNA) clusters in leaves and flowers.

## RdDM is recruited to CMT TEs independently of SHH1

The only H3K9me-binding factor implicated in Pol IV recruitment is SHH1 (*Law et al., 2013*; *Zhang et al., 2013*; *Zhou et al., 2018*). Therefore, we tested whether CMT TE hypermethylation in *h1* plants requires SHH1. CMT TEs remain hypermethylated in *h1cmt2shh1* plants to about the same extent as in *h1cmt2* plants (*Figure 4A*), demonstrating that in the absence of H1, Pol IV is recruited to CMT TEs independently of SHH1.

Pol IV activity depends on a family of four CLSY putative chromatin remodeling proteins (*Greenberg et al., 2013*; *Smith et al., 2007*; *Zhou et al., 2018*). Simultaneous loss of CLSY1 and CLSY2 has the same effect as loss of SHH1, whereas CLSY3 and CLSY4 mediate RdDM at a largely distinct set of loci (*Yang et al., 2018*; *Zhou et al., 2018*). Mutations of SHH1 and CLSY1/2 preferentially reduce mCHH and sRNA at DRM TEs and increase mCHH at CMT TEs (*Figure 4A*). In contrast, *clsy3/4* mutant plants have reduced mCHH and sRNA at CMT TEs and increased mCHH and sRNA at DRM TEs (*Figure 4A*), suggesting that SHH1 and CLSY1/2 preferentially mediate RdDM at DRM TEs, whereas CLSY3/4 preferentially recruit Pol IV to CMT TEs. Consistently, TEs hypermethylated in *h1cmt2* and *h1cmt2shh1* show a strong overlap with published CLSY3/4-dependent sRNA clusters and little overlap with CLSY1/2-dependent clusters (*Figure 4B* and *Figure 4—figure supplement 1*), suggesting that Pol IV relocation into heterochromatin involves CLSY3/4. However, our results do not rule out the possibility that some of the RdDM expansion in *h1* plants is mediated by CLSY1/2 or is independent of CLSY activity. Also, please note that the *wt* sRNA patterns in *Figures 1D and 4A* are distinct because the former is from leaves and the latter from inflorescences. Leaf sRNA levels are lower at CMT TEs and CLSY3/4 clusters compared to flowers (*Figure 4—figure supplement 2*), presumably due to higher expression of CLSY3/4 in reproductive tissues (*Long et al., 2021*; *Zhou et al., 2021*).

Overall, our results indicate that SHH1 is relatively unimportant for RdDM activity at H3K9me-rich CMT TEs with or without H1. The entry of Pol IV into H1-depleted heterochromatin must either involve a different H3K9me-interacting factor, or a chromatin feature other than H3K9me.

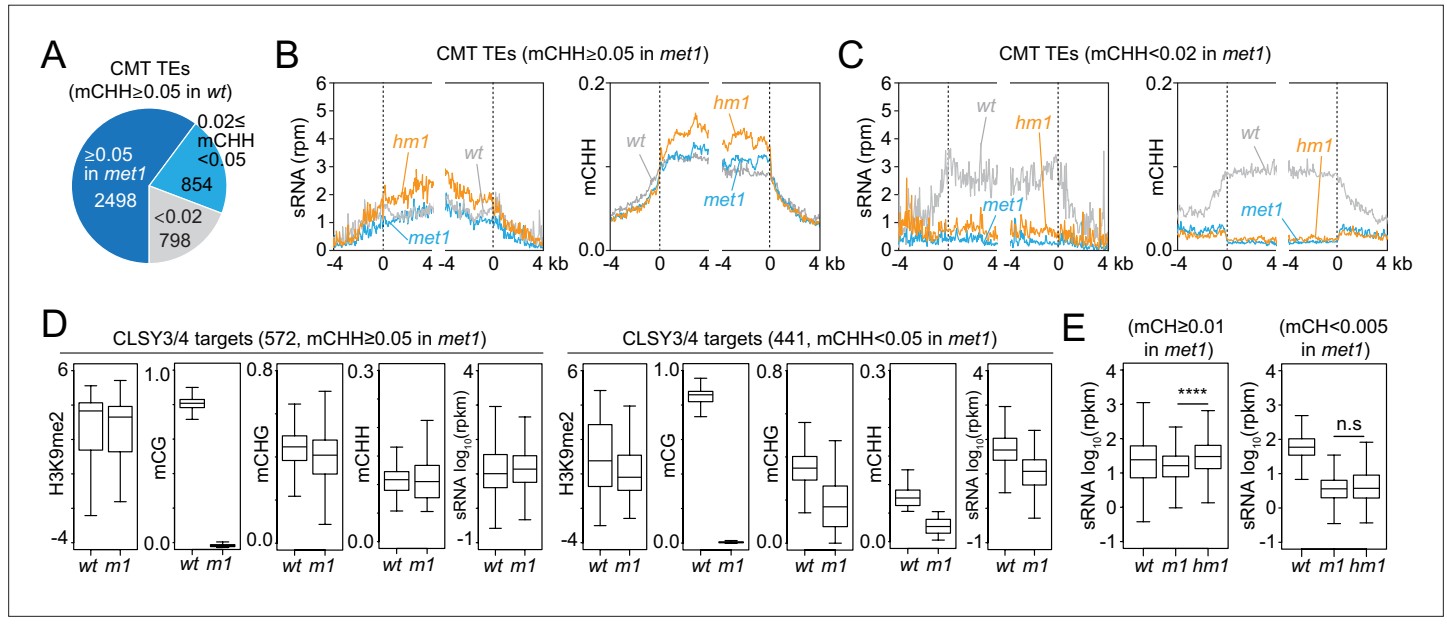

**Figure 5.** Small RNA (sRNA) expression at CLSY3/4 clusters is independent of CG methylation (mCG). (**A**) The number of chromomethylase (CMT) transposable elements (TEs; mCHH ≥0.05 in *wt*) that maintain mCHH in *met1* (mCHH ≥0.05 in *met1*; 2498) or lose mCHH in *met1* (mCHH <0.02 in *met1*; 798). (**B, C**) Averaged sRNA distribution and mCHH levels around CMT TEs in *wt*, *met1*, and *h1met1* (*hm*) plants that maintain mCHH in *met1* (mCHH ≥0.05 in *met1*; **B**) and lose mCHH in *met1* (mCHH <0.02 in *met1*; **C**). (**D**) Boxplots of H3K9me2, DNA methylation, and sRNA expression at CLSY3/4 sRNA clusters in *wt* and *met1* (*m1*). CLSY3/4 clusters that maintain more than 5% mCHH in *met1* or less than 5% mCHH in *met1* are plotted separately. (**E**) sRNA expression level at CLSY3/4 sRNA clusters that maintain non-CG methylation (mCH >0.01) in *met1* or lose non-CG methylation (mCH <0.005) in *met1*. Non-CG methylation (mCH) density equals number of mCH sites per base pair. **** indicates $p < 0.0001$.

The online version of this article includes the following figure supplement(s) for figure 5:

**Figure supplement 1.** DNA methylation and small RNA (sRNA) expression changes in *met1* and *h1met1*.

## RdDM expansion does not require mCG

Our results suggest that sRNA biogenesis at CMT TEs in *h1* mutants is mediated by CLSY3/4 Pol IV complexes. Recruitment of these complexes has been proposed to involve mCG (*Zhou et al., 2018*). Therefore, we examined sRNA levels and DNA methylation in *h1met1* mutants (*Choi et al., 2020*). Although MET1 is a CG methyltransferase, its loss also perturbs mCH and H3K9me2 at some CMT TEs (*Figure 5A* and *Figure 5—figure supplement 1A*; *Choi et al., 2020*; *Deleris et al., 2012*; *Zabet et al., 2017*; *Zhang et al., 2018a*). To understand how these changes impact sRNA production, we differentiate between two groups of CMT TEs in *met1* plants. MET1-independent CMT TEs keep mCH and H3K9me2 in *met1* (*Figure 5—figure supplement 1A*; *Choi et al., 2020*) and accordingly maintain sRNA expression (*Figure 5B*). These CMT TEs gain sRNA expression and mCHH in *h1met1* relative to *met1* and *wt* (*Figure 5B*), demonstrating that mCG is not required for RdDM expansion into heterochromatin. In contrast, MET1-dependent CMT TEs, which lose mCH and H3K9me in *met1* (*Figure 5—figure supplement 1A*; *Choi et al., 2020*), lose sRNA in *met1* and do not recover sRNA expression or mCHH in *h1met1* (*Figure 5C*), suggesting that mCH or H3K9me is necessary for sRNA biogenesis.

To test the above hypothesis, we grouped CLSY3/4 targets by mCHH level in *met1* (mCHH ≥0.05 in *wt* and *met1*; mCHH ≥0.05 in *wt* and <0.05 in *met1*). Even though all CLSY3/4 targets lose mCG in *met1*, sRNA expression is reduced only when mCH and H3K9me2 are reduced (*Figure 5D* and *Figure 5—figure supplement 1B*), implying that the presence of mCH and/or H3K9me is sufficient to maintain CLSY3/4-dependent sRNA biogenesis. In *h1met1*, sRNA levels increase at CLSY3/4 targets where mCH is maintained: among 1565 CLSY3/4 clusters with *wt* mCH (>0.01%), 72% keep mCH in *met1* and gain sRNA expression in *h1met1* (*met1* mCH >0.01), whereas 15% effectively lose all mCH in *met1* and have similarly low sRNA levels in *met1* and *h1met1* (*met1* mCH <0.005, *Figure 5E* and

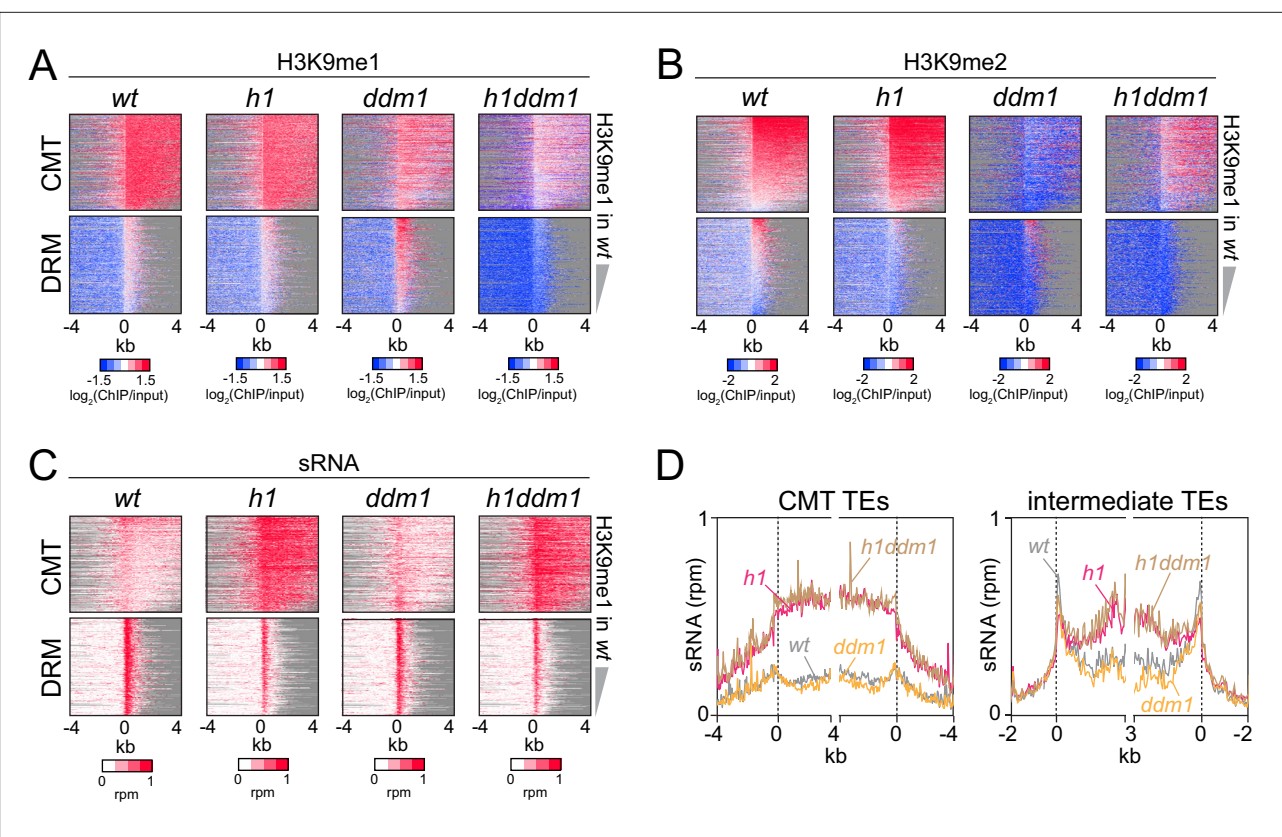

**Figure 6.** Severely reduced H3K9 methylation does not prevent small RNA (sRNA) expansion. (**A-C**) Distribution of H3K9 methylation (**A, B**) and sRNA expression (**C**) around 5' ends of chromomethylase (CMT) and DRM transposable elements (TEs) in *wt*, *h1*, *ddm1*, and *h1ddm1* plants. (**D**) Averaged sRNA distribution around CMT and intermediate TEs in *wt*, *h1*, *ddm1*, and *h1ddm1* plants.

The online version of this article includes the following figure supplement(s) for figure 6:

**Figure supplement 1.** Small RNA (sRNA) expression and H3K9 methylation changes in *h1*, *ddm1*, and *h1ddm1*.

*Figure 5—figure supplement 1C*). These results indicate that neither CLSY3/4 Pol IV activity, nor the RdDM expansion triggered by loss of H1, depend on mCG.

## Severe H3K9me reduction does not perturb RdDM expansion into heterochromatin

Our results so far indicate that H1 prevents RdDM from following a gradient of either H3K9me or mCH into heterochromatin. However, heterochromatin is structurally complex and contains many factors (*Feng and Michaels, 2015*). To understand the overall importance of heterochromatin integrity, we tested the effects of H1 on sRNA distribution in plants with a mutation in the Swi/Snf2 chromatin remodeler DDM1, which have severely compromised heterochromatin (*Kim and Zilberman, 2014*; *Sigman and Slotkin, 2016*). The *ddm1* mutation greatly reduces heterochromatic DNA and H3K9 methylation (*Ito et al., 2015*; *Lyons and Zilberman, 2017*; *Osakabe et al., 2021*; *Teixeira et al., 2009*; *Zemach et al., 2013*), activates TE expression (*Lippman et al., 2004*; *Osakabe et al., 2021*; *Panda et al., 2016*; *Panda and Slotkin, 2020*; *Rougée et al., 2021*), and disperses nuclear heterochromatic foci (*Rougée et al., 2021*; *Soppe et al., 2002*; *Figure 6A, B* and *Figure 6—figure supplement 1A*). However, 24-nt sRNA expression in *ddm1* is broadly similar to *wt* (*Figure 6C, D* and *Figure 6—figure supplement 1B*). Simultaneous lack of H1 and DDM1 in *h1ddm1* mutants (*Lyons and Zilberman, 2017*; *Zemach et al., 2013*) causes relocation of sRNA biogenesis into CMT and intermediate TEs that mirrors that in *h1* plants (*Figure 6C, D* and *Figure 6—figure supplement 1B*), indicating that overall heterochromatin integrity is not required for this process. Furthermore, RdDM expansion into heterochromatin occurs in *h1ddm1* despite strong H3K9me reduction compared to *wt* and *h1* (*Figure 6A, B* and *Figure 6—figure supplement 1A*). This does not rule out the possibility that H3K9me promotes Pol IV activity, because the H3K9me remaining in *h1ddm1* may be sufficient. However, the observation that sRNA production at CMT TEs is largely unaffected by a bulk H3K9me reduction argues against a primary role for H3K9me in Pol IV recruitment.

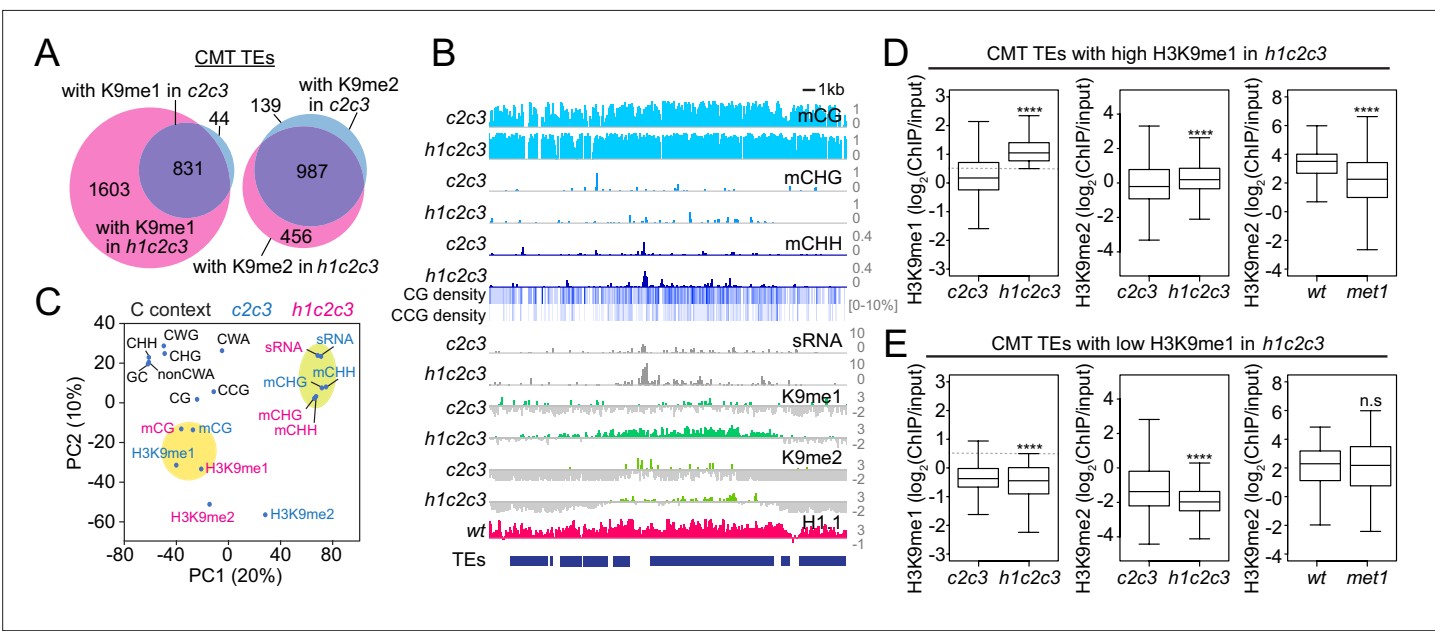

**Figure 7.** Non-CG DNA methylation and H3K9 methylation are decoupled in *h1c2c3*. (**A**) Number of chromomethylase (CMT) transposable elements (TEs) with H3K9 methylation (average H3K9me1 [K9me1] or H3K9me2 [K9me2] >0.5) in *cmt2cmt3* (*c2c3*) or *h1cmt2cmt3* (*h1c2c3*) plants. (**B**) Example of DNA methylation, CG and CCG density, H1 level, H3K9 methylation, and small RNA (sRNA) expression around CMT TEs in *c2c3* and *h1c2c3* plants (Chr3: 14,495,000–14,520,000). (**C**) Principal component analysis of H3K9me, cytosine content (total GC content, CG, CCG, CHG, CHH, CWG, CWA, and non-CWA [W = A and T]), DNA methylation, and sRNA expression in *c2c3* and *h1c2c3* plants. (**D, E**) H3K9me levels at CMT TEs with high H3K9me1 (H3K9me1 ≥0.5; **D**) or low H3K9me1 (H3K9me1 <0.5; **E**) in *h1c2c3* plants. **** indicates $p < 0.0001$.

The online version of this article includes the following figure supplement(s) for figure 7:

**Figure supplement 1.** Analysis of chromatin features at chromomethylase (CMT) transposable elements (TEs).

## H3K9me and mCH can be decoupled in heterochromatin

H3K9me and mCH are closely associated in heterochromatin due to the feedback loop between CMT2/3 and the SUVH4/5/6 H3K9 methyltransferases (*Du et al., 2012*; *Stoddard et al., 2019*; *Stroud et al., 2014*). To isolate the effects of these features on sRNA biogenesis, we examined DNA methylation, H3K9me and sRNA levels in *c2c3* and *h1c2c3* plants. While mCG is largely unaffected, mCH is specifically abolished at CMT TEs in these plants (*Figure 7—figure supplement 1A*), consistent with previously published *c2c3* results (*Stroud et al., 2014*). As expected, H3K9me is also greatly reduced (*Figure 7—figure supplement 1A*), but some H3K9me1 and H3K9me2 remains in heterochromatin. Specifically, 875 CMT TEs maintain H3K9me1 and 1126 maintain H3K9me2 in *c2c3*, while in *h1c2c3* we identified 2434 H3K9me1-enriched CMT TEs and 1443 H3K9me2-enriched CMT TEs (*Figure 7A, B*). Principal component analysis shows that H3K9me in these mutants associates with mCG, followed by CG and CCG density (which contribute to mCG density; *Figure 7C* and *Figure 7—figure supplement 1B*), suggesting that SUVH4/5/6 are recruited to mCG in the absence of mCH.

This conclusion is supported by a complementary pattern of H3K9 methylation changes in *h1c2c3* vs. *met1*. TEs that lose H3K9me2 in *met1*, suggesting H3K9me dependence on mCG, maintain H3K9me in the absence of mCH in *h1c2c3* (*Figure 7D*). Conversely, TEs that lose H3K9me in *h1c2c3*, suggesting H3K9me dependence on mCH, retain H3K9me2 in *met1* (*Figure 7E*). This indicates that H3K9me at mCG-dense CMT TEs is partially dependent on mCG, leading to considerable H3K9me retention in *c2c3*, and especially *h1c2c3* plants. The ability of mCG to recruit H3K9me is consistent with published work, including studies that show RdDM-independent initiation of the CMT-SUVH feedback loop specifically at CG-methylated sequences (*Miura et al., 2009*; *To et al., 2020*; *Zabet et al., 2017*) and the observed affinity of SUVH histone methyltransferase SRA domains for mCG in vitro (*Johnson et al., 2007*; *Li et al., 2018*; *Rajakumara et al., 2011*).

## 24-nt sRNA production specifically correlates with mCH

The decoupling of H3K9me and mCH in *h1c2c3* plants allowed us to determine how each feature is associated with sRNA biogenesis. In *h1* plants, H3K9me2, DNA methylation in every context, and sRNA expression together increase in direct relation to *wt* H1 prevalence, as loss of H1 increases accessibility of previously H1-rich TEs (*Figure 8A* and *Figure 8—figure supplement 1A*; *Bourguet et al., 2021*; *Lyons and Zilberman, 2017*; *Papareddy et al., 2020*; *Zemach et al., 2013*). H3K9me1/2, DNA methylation, and sRNA levels are also all positively correlated in *h1* plants, though the correlation between H3K9me2 and sRNA is weak (*Figure 8B* and *Figure 8—figure supplement 1B*). In contrast, the coupling of H3K9me with DNA methylation and sRNA levels nearly disappears when comparing *h1c2c3* to *c2c3* (*Figure 8C, D* and *Figure 8—figure supplement 1C, D*). Relative H3K9me1/2 abundance increases with *wt* H1 levels, whereas DNA methylation and sRNA changes show at best a very weak relationship with *wt* H1 enrichment (*Figure 8C* and *Figure 8—figure supplement 1C*).

Two correlated groups remain in *h1c2c3*: H3K9me1/2 with mCG, and sRNA with mCHG/mCHH (*Figure 8D* and *Figure 8—figure supplement 1D*). The linear correlations between sRNA and either H3K9me1 or mCG observed in *h1* (*Figure 8E*) become kinked in *h1c2c3* (*Figure 8F*), resembling the association between sRNA and H3K9me1 in *wt* (*Figure 3B*). The overall pattern of *h1c2c3* sRNA at CMT and intermediate TEs resembles *wt* far more than *h1* (*Figure 8G* and *Figure 8—figure supplement 1E*). The patterns and levels of sRNA and mCHH at DRM TEs are also similar between *h1c2c3* and *wt* (*Figure 8—figure supplement 1F, G*). Only the association between mCH and sRNA remains linear in *h1c2c3* (*Figure 8E, F*). This dynamic can be observed at an individual array of CMT TEs (*Figure 8H*). 24-nt sRNA expression is confined to the edges of the CMT TE array in *wt*, but follows H3K9me and DNA methylation throughout the array in *h1* plants (*Figure 8H*). In *h1c2c3*, mCH within the array is strongly reduced, but H3K9me is maintained, and sRNA expression exhibits a broadly *wt* pattern associated with remaining mCHH but not with H3K9me (*Figure 8H*).

It is important to note that in plants lacking CMT2/3, all mCHH should be catalyzed by RdDM, and a correlation between sRNA (product of the Pol IV pathway) and mCHH (product of the Pol V pathway) is therefore expected regardless of how Pol IV is recruited. The key observations are that loss of CMT2/3 in *h1c2c3* plants (and the associated loss of mCHG/mCHH) largely abrogates the relocation of Pol IV activity into heterochromatin (*Figure 8G,H* and *Figure 8—figure supplement 1E, G*), and the remaining heterochromatic sRNA biogenesis is not associated with H3K9me or mCG (*Figure 8D–F*). These results do not support the hypothesis that Pol IV is recruited by H3K9me, and

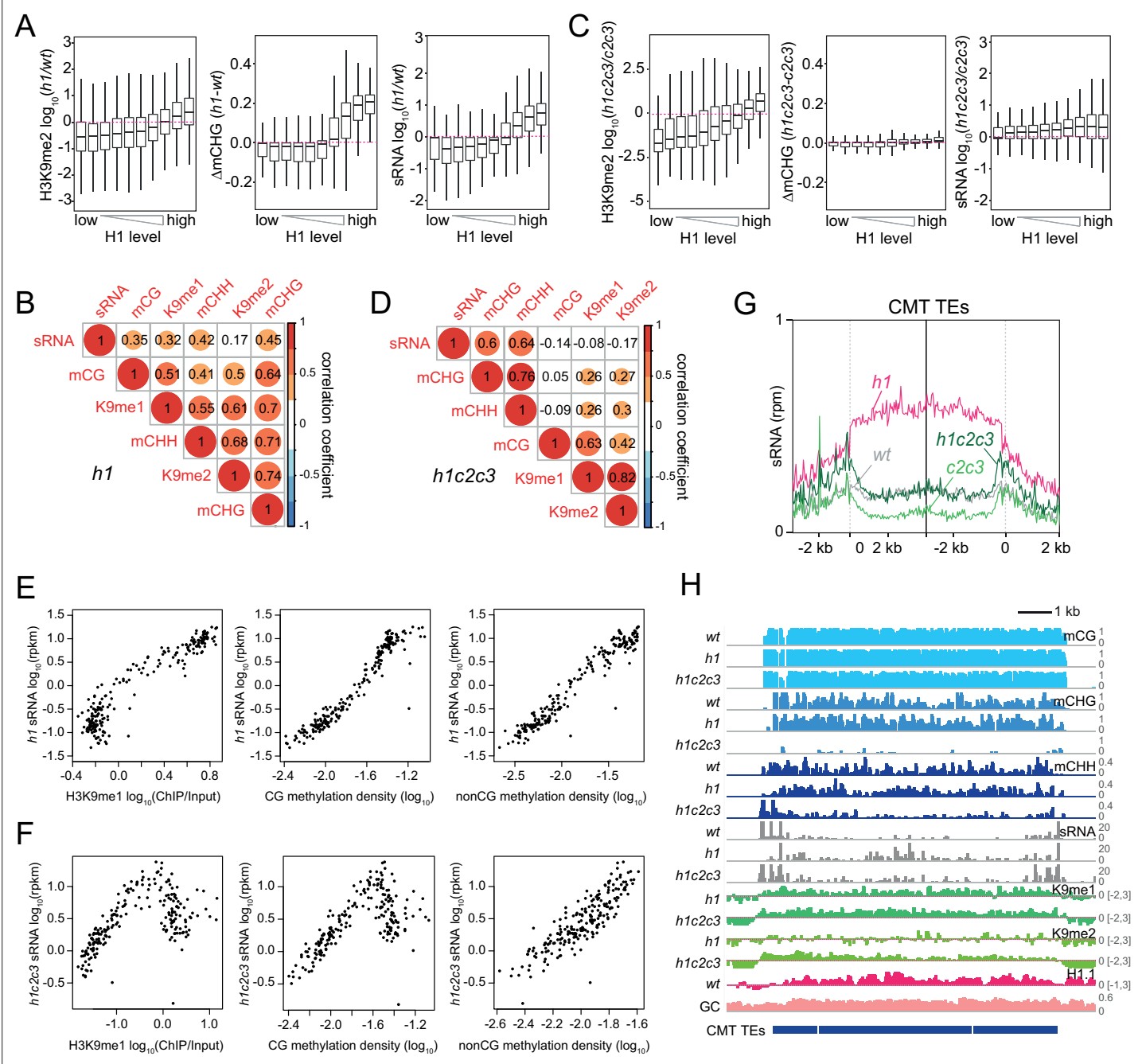

**Figure 8.** Small RNA (sRNA) expression specifically correlates with non-CG methylation. (A, C) Boxplots of H3K9me2, CHG methylation (mCHG), and sRNA expression changes in *h1* vs. *wt* (A) and *h1c2c3* vs. *c2c3* (C). (B, D) Correlation among H3K9 methylation, DNA methylation, and sRNA expression in *h1* plants (B) and *h1c2c3* plants (D). (E, F) sRNA expression relation to H3K9me1, CG, and non-CG methylation density in *h1* plants (E) and *h1c2c3* plants (F). Each dot represents the average of 100 transposable elements (TEs) sorted by GC content. DNA methylation density equals number of methylated sites per base pair. (G) Average sRNA expression level of chromomethylase (CMT) TEs in *wt*, *h1*, *c2c3*, and *h1c2c3* plants. (H) Example of DNA methylation, sRNA expression, H3K9 methylation (K9me1 and K9me2), and H1.1 distribution at CMT TEs in *wt*, *h1*, and *h1c2c3* plants (Chr2: 6,548,000–6,559,000).

The online version of this article includes the following figure supplement(s) for figure 8:

**Figure supplement 1.** H3K9 methylation, DNA methylation, and small RNA (sRNA) expression in *h1* and *h1c2c3* plants.

offer mCH as the most likely alternative. Our data suggest the hypothesis that without H1, mCH catalyzed by CMT2/3 pulls Pol IV into heterochromatin, and loss of CMT2/3 allows Pol IV to return to its mostly euchromatic *wt* targets.

## CLSY1/2 RdDM activity specifically associates with mCH

24-nt sRNA expression is globally associated with mCH rather than H3K9me in *h1c2c3*, but these correlations are primarily driven by heterochromatic regions with low *wt* RdDM. To determine if this trend translates to euchromatic TEs where SHH1 is required for RdDM, we analyzed associations between H3K9me, DNA methylation, and sRNA expression in published CLSY1/2 sRNA clusters in *wt* plants (*Figure 9A*; *Zhou et al., 2018*). In clusters grouped by H3K9me and mCHH, sRNA expression is associated with high mCHH, but not with high H3K9me (*Figure 9A*), supporting the idea that mCH dictates Pol IV localization (with the caveat that mCH is a product of RdDM).

As a further test of our hypothesis, we analyzed published data from plants lacking the three H3K9 methyltransferases implicated in the CMT/SUVH positive feedback loop. In these *suvh4/5/6* mutants, H3K9me2 and mCH are strongly diminished and sRNA expression of CLSY1/2 clusters is decreased (*Stroud et al., 2014*; *Zhou et al., 2018*). If H3K9me2 recruits Pol IV via SHH1, the limited remaining H3K9me would be expected to correlate with sRNA. Instead, we find sRNA expression in *suvh4/5/6* follows mCHH but not H3K9me2 (*Figure 9B, C*, compare left and right elements in *Figure 9C*), consistent with our observations in heterochromatin. 24-nt sRNA correlates much more strongly with mCH than with H3K9me2 in *suvh4/5/6* plants (*Figure 9D*), highlighting the limited importance of H3K9me for sRNA biogenesis.

Finally, we assayed CLSY1/2 clusters with low *wt* H3K9me2 but high *wt* sRNA and mCHH (LH CLSY1/2 clusters) in *polv* mutants to determine whether mCH is required to maintain sRNA expression. RNA Pol V is not directly involved in sRNA production, but is an essential RdDM component required for DNA methylation because it recruits DRM2 (*Erdmann and Picard, 2020*; *Matzke and Mosher, 2014*; *Raju et al., 2019*; *Wendte and Pikaard, 2017*). Therefore, *polv* mutants allow us to differentiate mCH as a cause vs. a consequence of Pol IV activity. 90% of the 662 LH CLSY1/2 clusters lose mCHH in *polv* plants (mCHH <0.05, *Figure 9E*), and the overall mCH of LH CLSY1/2 clusters is greatly reduced without Pol V (*Figure 9F*). In *suv4/5/6* mutants, LH CLSY1/2 clusters maintain sRNA expression, whereas sRNA expression in *polv* mutants is greatly reduced (*Figure 9G*). Furthermore, mCG at LH CLSY1/2 clusters is higher in *polv* than in *suvh4/5/6* plants (*Figure 9H*). Therefore, sRNA biogenesis is not sensitive to the loss of either H3K9me2 or mCG and specifically requires mCH.

## Discussion

We have examined intertwined chromatin features – sRNA production, DNA methylation, and H3K9 methylation – to understand how the genomic sites of Pol IV activity are specified. We find that two main factors are involved. First, linker histone H1 prevents sRNA production in heterochromatin (*Figure 10*). Without H1, RdDM relocates from its usual euchromatic targets into heterochromatic TEs (*Figure 1* and *Figure 1—figure supplement 1*), as has been recently observed by an independent study (*Papareddy et al., 2020*). Another heterochromatic protein, the histone variant H2A.W, may also contribute to the exclusion of RdDM from heterochromatin, but this effect is modest and only observed when H1 is absent (*Bourguet et al., 2021*). In the presence of H1, lack of H2A.W instead strengthens the exclusion of RdDM from heterochromatin, potentially due to enhanced heterochromatic H1 accumulation (*Bourguet et al., 2021*). Overall, the available evidence indicates that H1 is the major factor excluding Pol IV from heterochromatin.

Second, we find that mCH promotes Pol IV activity (*Figure 10*), contrary to the well-established view that Pol IV is recruited by H3K9me (*Erdmann and Picard, 2020*; *Law et al., 2013*; *Raju et al., 2019*; *Wendte and Pikaard, 2017*; *Zhang et al., 2013*), and the more recent proposal that mCG may be involved (*Zhou et al., 2018*). The hypothesis that mCH recruits Pol IV has a long history (*Herr et al., 2005*; *Li et al., 2020*; *Zemach et al., 2013*), but testing it has been challenging because mCH is associated with other epigenetic and chromatin features, including mCG and H3K9me (*Law and Jacobsen, 2010*; *Xu and Jiang, 2020*; *Zhang et al., 2018b*). The link with H3K9me has been particularly difficult to break because of the CMT-SUVH feedback loop (*Du et al., 2012*; *Johnson et al., 2007*; *Li et al., 2018*; *Stoddard et al., 2019*).

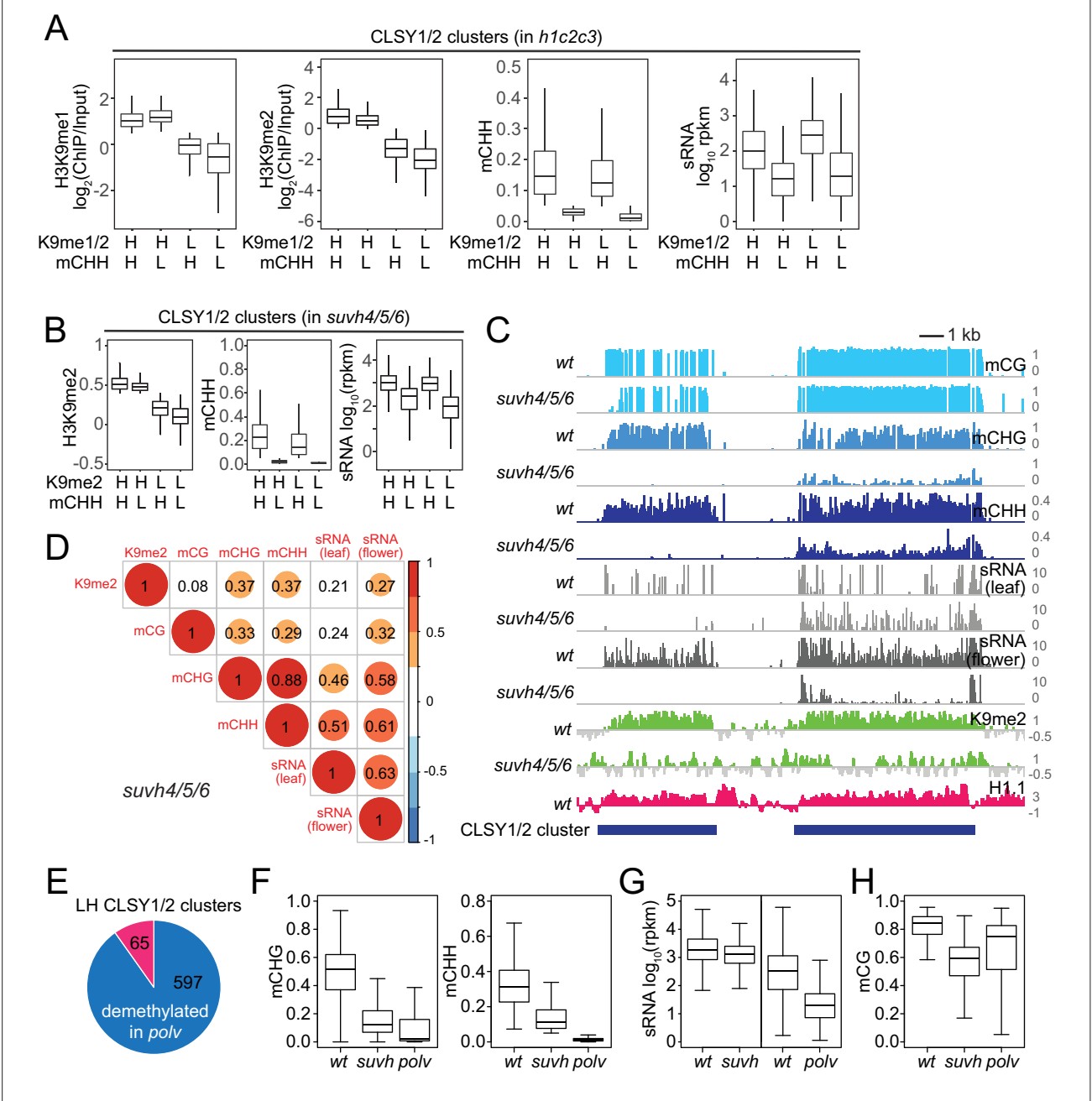

**Figure 9.** CLSY1/2-dependent small RNA (sRNA) expression is associated with non-CG methylation. (**A**) Boxplots of H3K9me1, H3K9me2, mCHH, and sRNA expression levels at CLSY1/2-dependent sRNA clusters in *h1c2c3* plants. sRNA clusters were classified by H3K9 methylation level (H3K9me1 >0.5, H3K9me2 >0 as high H3K9me [H], and the rest as low H3K9me [L]) and mCHH level (mCHH >0.05 as high mCHH [H] and the rest as low mCHH [L]). (**B**) Boxplots of H3K9me2, mCHH, and sRNA expression levels at CLSY1/2-dependent sRNA clusters in *suvh4/5/6* plants. sRNA clusters were classified by H3K9me2 level (H3K9me2 >0 as high H3K9me2 [H] and the rest as low H3K9me2 [L]) and mCHH level as in (**A**). (**C**) Examples of CLSY1/2 sRNA clusters with high H3K9me2 in *suvh4/5/6* but different non-CG methylation levels (Chr1: 17,520,000–17,538,000). (**D**) Correlation among H3K9me2, DNA methylation, and sRNA expression in *suvh4/5/6* plants. (**E**) Overlap between H3K9me2 low/mCHH high CLSY1/2 clusters (LH) in *suvh4/5/6* plants and mCHH demethylated CLSY1/2 clusters in *polv* plants. (**F-H**) Boxplots of non-CG methylation levels (**F**), sRNA expression (**G**), and mCG levels (**H**) at 597 CLSY1/2 clusters that lose mCHH in *polv* (blue in panel E).

However, we have used *h1c2c3*, *suvh4/5/6* and *polv* mutants to disentangle H3K9me and mCH. In all three backgrounds, sRNA biogenesis follows mCH instead of H3K9me (**Figures 8 and 9** and **Figure 8—figure supplement 1**). The *h1c2c3* line has been particularly informative due to the many TEs that maintain H3K9me but lack mCH (**Figure 8** and **Figure 8—figure supplement 1**). H3K9me

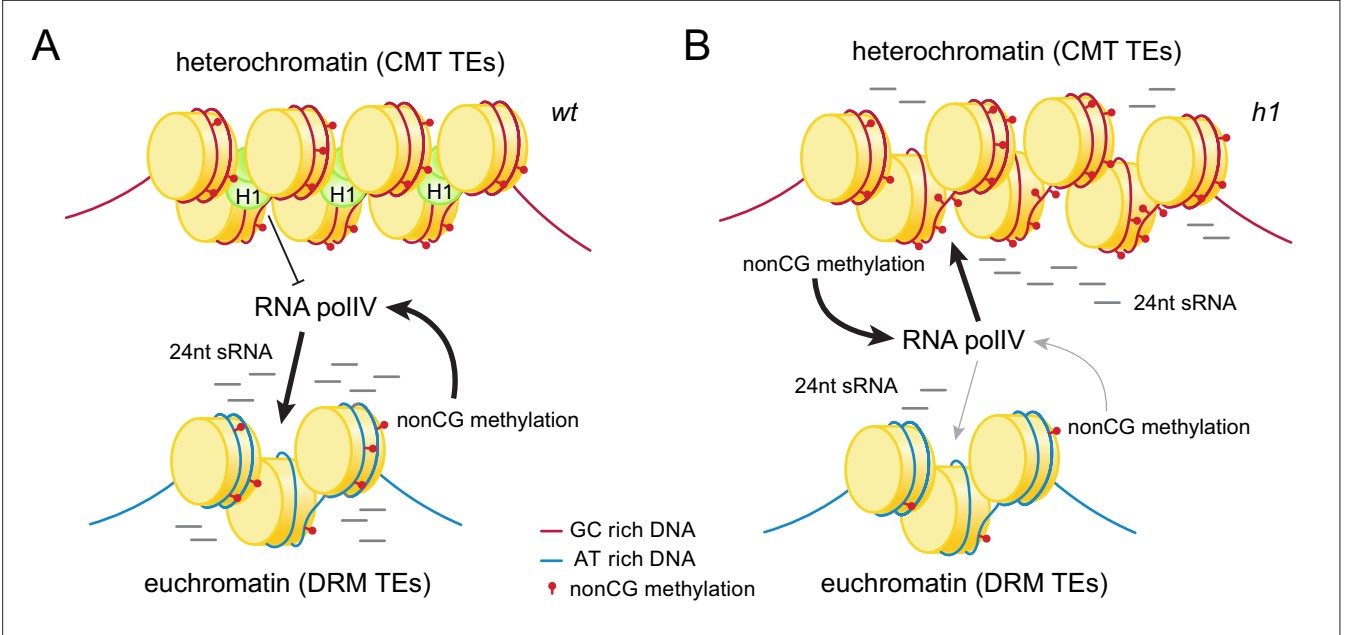

**Figure 10.** Histone H1 prevents non-CG methylation-mediated small RNA (sRNA) biogenesis in *Arabidopsis* heterochromatin. (**A**) In *wt* plants, H1 binds to GC-rich chromomethylase (CMT) transposable elements (TEs) to restrict access of RNA polymerase IV (Pol IV). Pol IV binds to DRM TEs and produces sRNA. (**B**) In *h1* plants, RNA Pol IV can transcribe non-CG-methylated CMT TEs to produce 24-nt sRNA, which leads to DNA methylation of CMT TEs and reduced activity at DRM TEs.

may be substantially retained in *h1c2c3* heterochromatin because lack of H1 allows SUVH methyltransferases easier access, so that the weak affinity of their SRA domains for mCG suffices for effective recruitment (*Johnson et al., 2007*; *Li et al., 2018*; *Rajakumara et al., 2011*). Whatever the mechanism, the strong linear association between sRNA biogenesis and mCH, and the lack of such an association with H3K9me and mCG (*Figures 8 and 9*), provide strong support for the hypothesis that mCH recruits Pol IV (*Figure 10*).

Our data linking 24-nt biogenesis with mCH do not mean that such methylation is absolutely required for Pol IV recruitment. Indeed, there is residual 24-nt biogenesis in *ddcc* mutants that lack mCH (*Stroud et al., 2014*). One possibility is that the factor or factors recruiting Pol IV to mCH have weak affinity for mCG, which could recruit Pol IV in the absence of mCH, analogous to our proposed mode of SUVH4/5/6 recruitment in plants lacking CMT2/3. Other chromatin features may also recruit or facilitate Pol IV activity. However, our results indicate that mCH is the major Pol IV recruiting genomic feature under normal conditions.

The linking of Pol IV activity to mCH instead of H3K9me resolves several thorny issues. First, the observation that SHH1 – the proposed H3K9me reader – is preferentially required for RdDM where H3K9me is low (*Zhou et al., 2018*) can be easily accommodated if H3K9me is not directly involved in RdDM. Similarly, the finding that severe loss of H3K9me in *suvh4/5/6* mutants is accompanied by only a modest reduction of sRNA levels (*Zhou et al., 2018*) is no longer mysterious. At a more fundamental level, this hypothesis ties RdDM in a feedback loop with its product and unties it from a histone modification produced by the distinct CMT-SUVH pathway and depleted from RdDM target sequences. Breaking RdDM from dependence on any histone modification is also conceptually important because a core theoretical strength of RdDM is the ability to maintain methylation at much shorter sequences than those where stable histone-based epigenetic inheritance is possible (*Angel et al., 2011*; *Lövkvist and Howard, 2021*; *Ramachandran and Henikoff, 2015*; *Zilberman and Henikoff, 2004*).

Long TEs that can be effectively silenced by the histone-dependent CMT-SUVH pathway tend to be relatively GC-rich because they contain coding sequences (*Sequeira-Mendes et al., 2014*; *To et al., 2020*; *Zemach et al., 2013*). In contrast, short nonautonomous TEs and TE remnants tend to lack coding sequences and are thus AT-rich. In this context, the GC sequence preference of *Arabidopsis* H1 (*Choi et al., 2020*) may be key. GC bias is far from a H1 universal, with most animal H1 variants preferring AT-rich DNA (*Cao et al., 2013*; *Izaurralde et al., 1989*; *Tomaszewski and Jerzmanowski, 1997*).

The preferences of plant H1 may have evolved, at least in part, to target it to coding sequences, including those of autonomous heterochromatic TEs. This would allow H1 to exclude RdDM from such sequences, which can cover vast tracts of plant genomes (*Michael, 2014*; *Suzuki and Bird, 2008*), and focus RdDM on the short TEs it is specialized to silence. The interplay of H1 and mCH can thus produce the preferential activity of RdDM at short, AT-rich TEs observed throughout flowering plants (*Gouil and Baulcombe, 2016*; *Numa et al., 2015*; *Tan et al., 2018*).

# Materials and methods

**Key resources table**

| Reagent type (species) or resource | Designation | Source or reference | Identifiers | Additional information |
|---|---|---|---|---|
| Antibody | anti-H3K9me1 (Rabbit polyclonal) | Millipore | 07-450 RRID:AB_310625 | 1:200 |
| Antibody | anti-H3K9me2 (Mouse monoclonal) | Abcam | ab1220 RRID:AB_449854 | 1:200 |
| Commercial assay or kit | Library construction (Native ChIP) | Tecan | 3460-24 | |
| Commercial assay or kit | Bisulfite conversion | QIAGEN | 59,104 | |
| Commercial assay or kit | Library construction (bisulfite sequencing) | New England Biolabs | E7645 and E7335S | |
| Commercial assay or kit | Library construction (small RNA) | Illumina | RS-200-0012 and RS-200-0024 | |
| Software, algorithm | cutadapt | doi:10.14806/ej.17.1.200 | RRID:SCR_011841 | |
| Software, algorithm | bowtie | doi:https://doi.org.10.1186/gb-2009-10-3-r25 | RRID:SCR_005476 | |
| Software, algorithm | deepTools2 | doi:10.1093/nar/gkw257 | | |
| Software, algorithm | dzlabtools | doi:10.1126/science.1172417 | | https://zilbermanlab.net/tools/ |
| Software, algorithm | RandomForestExplainer | doi:10.1198/jasa.2009.tm08622 | | |
| Software, algorithm | IGV | doi:10.1038/nbt.1754 | RRID:SCR_011793 | |
| Software, algorithm | Gene Cluster 3.0 | doi:10.1093/bioinformatics/bth078 | | |
| Software, algorithm | corrplot | doi:10.1198/000313002533 doi:10.1080/00031305.1996.10474371 | | |
| Software, algorithm | Treeview | doi:10.1093/bioinformatics/bth078 | RRID:SCR_016916 | |

## Biological materials

*cmt2* and *cmt2cmt3* (*Stroud et al., 2014*; *Zemach et al., 2013*) plants were crossed to *h1.1h1.2* (*Zemach et al., 2013*) plants to generate *h1cmt2* and *h1cmt2cmt3* plants. To establish the *h1cmt2shh1* mutant line, we crossed *h1 +/- cmt2* plants with *shh1* (SALK_074540C) plants, then isolated *h1cmt2shh1* homozygous siblings. *met1*, *h1met1*, *ddm1*, and *h1ddm1* plants were described previously (*Choi et al., 2020*; *Lyons and Zilberman, 2017*). *Arabidopsis thaliana* seedlings were germinated and grown for 4–5 weeks on soil at 20–25°C in growth chambers (16 hr day/8 hr night) for all the experiments performed except for *met1*, *h1met1*, and corresponding *wt* seedling sRNA libraries. These seedlings were germinated and grown for 2 weeks in half-strength Gamborg's B-5 liquid media (Caisson Labs, cat. no. GBP07) at 22–25°C under continuous light with shaking at 125 rpm.

## Bisulfite sequencing library preparation

Bisulfite sequencing (BS-seq) libraries were constructed using genomic DNA (gDNA) extracted from rosette leaves of 4–5-week-old plants. 500 ng total gDNA was sheared to 100–1000 bp using Bioruptor

Pico (Diagenode), then purified with 1.2× volume of SPRI beads (Beckman Coulter, cat. no. A63881). Fragmented gDNA was ligated to NEBNext Adaptor for Illumina using NEBNext Ultra II DNA library prep kit for Illumina (New England Biolabs, cat. no. E7645). We performed bisulfite conversion twice with ligated libraries (QIAGEN, cat. no. 59104) to prevent incomplete conversion (<99% conversion) of unmethylated cytosines. Converted libraries were subjected to SPRI bead purification with 0.8× volume of beads. We amplified bisulfite-converted libraries with NEB next indexing primers (New England Biolabs Inc, cat. no. E7335S).

## sRNA-sequencing library preparation
To isolate sRNA, we extracted total RNA from rosette leaves of 4–5-week-old plants using Trizol (Invitrogen, cat. no. 15596026) according to the manufacturer's manual. To remove DNA from samples, 5 µg of RNA was treated with DNA-free DNA removal kit (Thermo, cat. no. AM1907). 1 µg of DNA-free total RNA was subjected to sRNA library construction according to the manufacturer's protocol (Illumina, cat. no. RS-200-0012 and RS-200-0024).

## Native chromatin immunoprecipitation and sequencing library preparation
MNase digestion of native chromatin was carried out on 0.5 g of 4-week-old *Arabidopsis* rosette leaves as described previously (*Lyons and Zilberman, 2017*). Digestion was stopped with EGTA and chromatin was rotated at 4°C for 30 min. The preparation was then centrifuged for 10 min at 2000 rpm and solubilized chromatin fragments were isolated by aspirating supernatant immediately. Chromatin was then diluted to 1 ml in wash buffer A (50 mM Tris–HCl pH 7.5, 50 mM NaCl, 10 mM EDTA) and antibody added at 1 µl per 0.1 g of total starting material (Millipore, cat. no. 07-450 for H3K9me1, Abcam, cat. no. ab1220 for H3K9me2). Dilute Tween-20 was added to a final concentration of 0.1%, and the mixture was rotated overnight at 4°C. All buffers were supplemented with PMSF and protease inhibitor (Roche [Merck], cat. no. 11873580001). A standard immunoprecipitation procedure was used the following day. Briefly, preblocked Protein-A and -G dynabeads (Invitrogen, cat. no. 10,001D and 10,003D) were incubated with the chromatin preparation for 3 hr. rotating at 4°C, and the beads/chromatin mixture was then washed on ice in Tris–EDTA buffer with increasing concentrations of NaCl, starting at 50 mM and ending at 150 mM. DNA was eluted from beads by shaking in 1% SDS and 1% NaHCO₃ for 10 min at 55°C, and DNA was purified with phenol–chloroform extraction. Input and ChIP DNA was converted into sequencing libraries using Celero DNA reagents (Tecan, cat. no. 3460-24) following the manufacturer's instructions.

## Sequencing
Sequencing was performed at the John Innes Centre with the NextSeq 500 (Ilumina), except for sRNA libraries from seedlings (*wt*, *met1*, and *h1met1*). These seedling libraries were sequenced at the Vincent J. Coates Genomic Sequencing Laboratory at the University of California, Berkeley with the HiSeq 4000 (Illumina).

## Sequence alignment and data preparation
For sRNA-seq libraries, adapter sequences were removed from reads using cutadapt (*Martin, 2011*). 18–28 bp, 21 nt, and 24 nt fragments were isolated using the following cutadapt options: -m 18 M 28, -m 21 M 21, -m 24 M 24. Reads were mapped with Bowtie (*Langmead et al., 2009*) allowing up to one mismatch and up to 10 multimapped reads. Aligned 21-nt or 24-nt read counts were normalized by reads per kilobase per million mapped reads (rpkm) of 18–28 bp fragments. ChIP-seq libraries were mapped with Bowtie (*Langmead et al., 2009*) allowing up to 2 mismatches and up to 10 multimapped reads. To calculate enrichment, ChIP samples were divided by input samples and transformed into $\log_2$ ratio values using deepTools2 bamCompare (*Ramírez et al., 2016*). For H3K9me1 and H3K9me2 from WT, *h1*, *ddm1*, *h1ddm1*, *c2c3*, and *h1c2c3*, we used a random subset of input reads equivalent to 25% of the total uniquely mapped reads of the corresponding IP for input into bamCompare. For BS-seq libraries, reads were mapped with the bs-sequel pipeline (https://zilbermanlab.net/tools/).

## Description of *Arabidopsis* genome features

'Transposable elements' include transposon annotation from *Panda and Slotkin, 2020*. Araport11 TE genes and pseudogenes, and genomic regions with TE-like DNA methylation (*Cheng et al., 2017*; *Choi et al., 2020*; *Panda and Slotkin, 2020*; *Shahzad et al., 2021*). We filtered out elements shorter than 250 bp. Previously, we merged overlapping TE annotations into single TE unit, then defined heterochromatic TEs and euchromatic TEs as transposons that have more than 0 or less than 0 H3K9me2 ($\log_2$ ChIP/Input) in *wt* plants (*Choi et al., 2020*). Both CMT and DRMs target these merged, long TEs, as the edges of TEs are methylated by DRMs and the bodies of TEs are methylated by CMTs. Therefore, to isolate TEs with mCH dependent on CMTs or DRMs, we did not merge TE annotations here. Among TEs with mCHH methylation (mCHH >0.02), CMT-dependent TEs were defined as the TEs that lost mCHH methylation in *cmt2* plants (mCHH <0.02 in *cmt2*). DRM-dependent TEs were defined as the TEs that lost mCHH methylation in *drm2* plants (mCHH <0.02 in *drm2*). sRNA cluster annotation is from *Zhou et al., 2018*.

## Classification of MET1-dependent and -independent CMT TEs

We previously defined MET1-dependent TEs as the TEs that lost H3K9me2 in *met1* plants (*Choi et al., 2020*). In this study, to evaluate how DNA methylation affects CLSY3/4-dependent sRNA expression, we defined MET1-dependent TEs as the TEs that lost mCHH methylation in *met1* (mCHH in *wt* ≧0.05, mCHH in *met1* <0.02), and MET1-independent TEs as ones that keep mCHH methylation in *met1* (mCHH in *wt* ≧0.05, mCHH in *met1* ≧0.05).

## Random forest classification and prediction

To measure the importance of each genetic and epigenetic marker to classify DRM and CMT TEs, we first calculated average enrichment of various histone modifications, histone H1, average sRNA expression, and DNA methylation level at each TE using window_by_annotation.pl Perl script (https://zilbermanlab.net/tools/). We also included density of various cytosine sequence contexts. The importance of each variable was evaluated using 'randomForest' and 'measure_importance' function in RandomForestExplainer R package (*Ishwaran et al., 2012*). The importance matrices were visualized by 'plot_multi_way_importance' function of the same package.

To evaluate the predictive power of each variable, we randomly divided TEs into training and validation sets. The random forest classifier was built using TEs in the training set with indicated variables and the classification of each TE (DRM or CMT). The trained model was used to predict the category of TEs in the validation set, and the error rate was calculated by comparing the predicted classification and its actual classification. We used 'randomforest' and 'predict' function in randomForest R package.

## Data visualization

Enrichment scores of various genomic and epigenomic features were generated by window_by_annotation.pl Perl scripts (https://zilbermanlab.net/tools/). For scatter plots and heatscatter plots in *Figure 1*, the enrichment scores were imported to R (*Davey et al., 1997*) and visualized by ggplot2 R package (*Wickham, 2009*) or 'heatscatter' function in LSD R package (*Venables and Ripley, 2002*). For scatter plots and heatscatter plots in other figures, TEs were sorted by their GC content, then average feature enrichments of 100 TEs were calculated to reduce the variability of data. DNA methylation, H3K9 methylation, and sRNA distribution around TEs were generated with ends_analysis.pl and average_ends_new.pl Perl scripts (https://zilbermanlab.net/tools/). For sRNA distribution, we removed bins with higher than 200 rpkm to prevent outliers skewing the average. For proportional Venn diagram, TE ID lists in each group were uploaded to BioVenn (*Hulsen et al., 2008*). To visualize the relationship among genetic, epigenetic features and sRNA expression in *c2c3* and *h1c2c3* plants, principal component analysis was applied to arrays of features using Gene Cluster 3.0 (*de Hoon et al., 2004*; *Figure 6C*). For Pearson's correlation coefficient plots, the DNA methylation, H3K9 methylation, and sRNA expression level matrices were imported to R and visualized using corrplot R package (*Friendly, 2002*; *Murdoch and Chow, 1996*; *Figures 7 and 8*). Screenshots of *Arabidopsis* genomic loci were taken in IGV (*Robinson et al., 2011*; *Thorvaldsdóttir et al., 2013*). Treeview was used to generate heatmaps (*de Hoon et al., 2004*). For sRNA plots around nucleosomes (*Figure 2*), previously published nucleosome dyad coordinates were used (*Lyons and Zilberman, 2017*) as anchors

around which 10 bp bins of 24-nt sRNA were averaged and plotted. Autocorrelation estimates were generated on these averages using the built-in R 'acf' function.

## Use of previously published data

DNA methylation data of *wt*, *drm2*, *c2c3*, *ddcc*, and *ibm1* plants (*Stroud et al., 2014*; *Zemach et al., 2013*), DNA methylation and sRNA data of *clsy1/2*, *clsy3/4*, and *shh1* plants (*Zhou et al., 2018*), DNA methylation, MNase, well-positioned nucleosome loci data of *wt* and *h1* plants (*Lyons and Zilberman, 2017*), DNA methylation, H1 and H3K9me data of *wt*, *met1*, and *h1met1* plants (*Choi et al., 2020*), H3K9me2 and sRNA expression data of *wt* and *ibm1* plants (*Fan et al., 2012*; *Lai et al., 2020*), DNA methylation, H3K9me2, and sRNA expression data of *suvh4/5/6* plants (*Papareddy et al., 2020*; *Stroud et al., 2014*), and DNA methylation and sRNA data of *polv* plants (*Johnson et al., 2014*; *Zhong et al., 2012*) were obtained through GEO (GEO accessions: GSE51304, GSE41302, GSE99694, GSE122394, GSE108487, GSE32284, GSE152971, GSE52041, and GSE39247).

## Acknowledgements

We thank X Feng for helpful comments on the manuscript. This work was supported by a European Research Council grant MaintainMeth (725746) to DZ.

# Additional information

### Competing interests

Daniel Zilberman: Reviewing editor, *eLife*. The other authors declare that no competing interests exist.

### Funding

| Funder | Grant reference number | Author |
| --- | --- | --- |
| H2020 European Research Council | 725746 | Choi Jaemyung<br>Lyons David B<br>Daniel Zilberman |

The funders had no role in study design, data collection, and interpretation, or the decision to submit the work for publication.

### Author contributions

Jaemyung Choi, Formal analysis, Investigation, Methodology, Writing – original draft, Writing – review and editing, Conceptualization; David B Lyons, Formal analysis, Investigation, Methodology, Writing – original draft, Writing – review and editing; Daniel Zilberman, Conceptualization, Funding acquisition, Project administration, Supervision, Writing – review and editing

### Author ORCIDs

Jaemyung Choi http://orcid.org/0000-0002-5725-404X
David B Lyons http://orcid.org/0000-0002-5721-4080
Daniel Zilberman http://orcid.org/0000-0002-0123-8649

### Decision letter and Author response

Decision letter https://doi.org/10.7554/eLife.72676.sa1
Author response https://doi.org/10.7554/eLife.72676.sa2

# Additional files

### Supplementary files

• Transparent reporting form

### Data availability

Sequencing data have been deposited in GEO under accession code GSE179796.

The following dataset was generated:

| Author(s) | Year | Dataset title | Dataset URL | Database and Identifier |
|---|---|---|---|---|
| Choi J, Lyons DB | 2021 | Histone H1 prevents non-CG methylation-mediated small RNA biogenesis in Arabidopsis heterochromatin | https://www.ncbi.nlm.nih.gov/geo/query/acc.cgi?acc=GSE179796 | NCBI Gene Expression Omnibus, GSE179796 |

The following previously published datasets were used:

| Author(s) | Year | Dataset title | Dataset URL | Database and Identifier |
|---|---|---|---|---|
| Stroud H | 2013 | Non-CG methylation patterns shape the epigenetic landscape in Arabidopsis | https://www.ncbi.nlm.nih.gov/geo/query/acc.cgi?acc=GSE51304 | NCBI Gene Expression Omnibus, GSE51304 |
| Zemach A, Hsieh P, Coleman-Derr D, Thao K, Harmer SL, Zilberman D | 2013 | DDM1 and RdDM are the major regulators of transposon DNA methylation in Arabidopsis | https://www.ncbi.nlm.nih.gov/geo/query/acc.cgi?acc=GSE41302 | NCBI Gene Expression Omnibus, GSE41302 |
| Zhou M, Palanca AMS, Law JA | 2018 | Locus-specific control of the de novo DNA methylation pathway | https://www.ncbi.nlm.nih.gov/geo/query/acc.cgi?acc=GSE99694 | NCBI Gene Expression Omnibus, GSE99694 |
| Choi J, Lyons DB, Kim MY | 2019 | DNA methylation and histone H1 jointly repress transposable elements and aberrant intragenic transcripts | https://www.ncbi.nlm.nih.gov/geo/query/acc.cgi?acc=GSE122394 | NCBI Gene Expression Omnibus, GSE122394 |
| Lai Y, XM Lu, Le Roche K, Eulgem T | 2020 | Genome-wide profilings of EDM2-mediated effects on H3K9me2 and transcripts in Arabidopsis thaliana | https://www.ncbi.nlm.nih.gov/geo/query/acc.cgi?acc=GSE108487 | NCBI Gene Expression Omnibus, GSE108487 |
| Fan D, Wang X, Zhang J, Ma L | 2012 | IBM1, a JmjC domain histone demethylase, is involved in the regulation of RNA-directed DNA methylation through epigenetic control of RDR2 and DCL3 expression in Arabidopsis. | https://www.ncbi.nlm.nih.gov/geo/query/acc.cgi?acc=GSE32284 | NCBI Gene Expression Omnibus, GSE32284 |
| Nodine M, Papareddy R | 2020 | Chromatin regulates expression of small RNAs to help maintain transposon methylome homeostasis in Arabidopsis | https://www.ncbi.nlm.nih.gov/geo/query/acc.cgi?acc=GSE152971 | NCBI Gene Expression Omnibus, GSE152971 |
| Hale CJ | 2014 | SRA/SET domain-containing proteins link RNA polymerase V occupancy to DNA methylation | https://www.ncbi.nlm.nih.gov/geo/query/acc.cgi?acc=GSE52041 | NCBI Gene Expression Omnibus, GSE52041 |
| Hale CJ | 2012 | The DDR complex facilitates the genome-wide association of RNA Polymerase V to promoters and evolutionarily young transposons | https://www.ncbi.nlm.nih.gov/geo/query/acc.cgi?acc=GSE39247 | NCBI Gene Expression Omnibus, GSE39247 |

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
