## [Decision Letter]

**Decision letter after peer review:**

Thank you for submitting your article "Histone H1 prevents non-CG methylation-mediated small RNA biogenesis in Arabidopsis heterochromatin" for consideration by *eLife*. Your article has been reviewed by 3 peer reviewers, and the evaluation has been overseen by Reviewing Editor Rick Amasino and Detlef Weigel as the Senior Editor. The following individuals involved in review of your submission have agreed to reveal their identity: Craig S Pikaard (Reviewer #1); Michael D Nodine (Reviewer #3).

The reviewers have discussed their reviews with one another, and the Reviewing Editor has drafted the following note help you prepare a revised submission.

The reviewers and editors agree that this is interesting work and although in the reviews there are suggestions for some experiments that might be done, further experiments are not required. There are however many points in the thoughtful and thorough reviews that ought to be incorporated into the revised version to make it stronger contribution to the field. We look forward to receiving your revised manuscript.

*Reviewer #1:*

This paper examines RNA-directed DNA methylation (RdDM) at transposable elements (TEs) in wild-type plants versus plants with defective mutations in genes encoding linker histone H1, the CG DNA methyltransferase MET1, the CHG and CHH DNA methyltransferases CMT2 and CMT3, the de novo DNA methyltransferases DRM1 and DRM2, the chromatin remodeling enzyme DDM1, the putative chromatin remodeling proteins CLSY1, CLSY2, CLSY3 and CLSY4, the histone H3 lysine 9 methyl group (H3K9me) binding protein SHH1, the Histone H3 lysine 9 methyltransferases SUVH4,5 and 6 and the DNA-dependent RNA polymerases Pol IV and Pol V. This is an extensive list of mutants, which are examined in a number of new combinations. Mutations combined with the loss of histone H1 yielded several new insights. Without histone H1, TEs whose methylation primarily depends on CMT2/3 (CMT TEs) were found to gain methylation whereas TEs that are mainly methylated by DRM1/2 became hypomethylated. H3K9me levels are typically high at DNA loci with high levels of cytosine methylation, and SHH1 is a protein that can bind H3K9me and is implicated in recruitment of Pol IV to initiate the small RNA (sRNA) part of the RdDM pathway. Contrary to current thinking, the authors argue that H3K9 methylation binding by SHH1 plays a small role in recruiting RdDM machinery to CMT TEs in h1 mutants, as loss of SHH1 did not diminish an increase sRNAs and CHH hypermethylation at CMT TEs. The authors present evidence that sRNA biogenesis at CMT TEs in h1 mutants requires CLSY3/4, as a large proportion of hypermethylated TEs in h1cmt2 and h1cmt2shh1 mutants overlaps with the CLSY3/4-dependent sRNAs. To test if CLSY3/4 recruitment is dependent on CG methylation, as previously proposed, a h1met1 line was used. The authors found that the loss of CG methylation and histone H1 in h1met1 mutants resulted in an increase in methylation at CHH motifs instead of the expected decrease, suggesting that CG methylation is not the chromatin mark key to CLSY3/4-dependent RdDM detected at heterochromatic TEs in h1 mutants. The authors propose instead that methylated mCHG/mCHH motifs are the primary marks for recruiting the RdDM machinery, with H3K9me-dependent mechanisms (e.g. via SHH1 and CLSY1/2) playing a lesser role.

Many of the claims are supported by the data presented and are convincing; however, some claims, including the conclusion, need more data or further clarification.

Primary issues to be addressed:

1. The authors hypothesize that histone H1 prohibits RdDM from occurring at CMT2-dependent heterochromatic regions, which have high levels of H1 association. This is supported by the observation that plants with mutations in H1 genes (h1) exhibit elevated mCHH levels at these loci. In contrast, histone H1 is depleted in DRM-dependent regions (Figure 1—figure supplement 1), which is hypothesized to allow RdDM to occur. One might expect mCHH levels to increase at DRM-dependent regions in h1 mutations, but CHH motifs at the DRM-dependent regions become hypomethylated. The authors speculate that dilution of RdDM machinery by expanding to CMT regions may contribute to the reduced mCHH at the DRM TEs. However, the 'dilution' hypothesis is highly speculative and not supported by direct evidence. The "dilution" idea brought to mind a 2012 study by Wierzbicki et al., that found that total levels of CHH methylation remained fairly constant when wild-type plants, pol IV mutant plants or pol V mutant plants were compared, but with CHH methylation increasing in pericentromeric regions and decreasing in chromosome arms in the pol IV/V mutants. An increase in CHH methylation in pericentric regions due to increased RdDM seems unlikely as an explanation for the 2012 results, as RdDM is expected to be eliminated in pol IV and pol V mutants. Instead, the pericentromeric methylation is presumably attributable to CMT2 activity based on the current and former studies of the Zilberman lab. Collectively, these old and new results suggests that CMT2 and RdDM-dependent methylation may involve competition for a common factor present in limiting concentration. As an example, one candidate might be the methyl donor S-adenosyl methionine (AdoMet). If there is competition for a limiting substrate, knocking out either the CMT or DRM-dependent methylation pathway may simply make more AdoMet available to the other pathway, resulting in higher levels of methylation. The authors should consider such alternative hypothesis, as opposed to the spreading of the machinery for one pathway into new territories. Conducting ChIP experiments would also allow tests for the authors' spreading hypothesis.

2. One major conclusion in the manuscript is that RdDM/Pol IV is recruited to non-CG methylated loci regardless of H3K9 methylation. Multiple observations support this conclusion: (a). CMT TEs are similarly hypermethylated in h1cmt2shh1 and h1cmt2 plants (Figure 3C), suggesting that the CHH methylation at the CMT regions are mediated independently of SHH1, which is a H3K9me binding protein thought to mediate Pol IV recruitment; (b). sRNAs hyper-accumulate in a h1ddm1 mutant despite a large decrease in H3K9me in this background (Figure 5); (c). sRNA accumulation better correlates with non-CG methylation than with H3K9me in the h1cmt2cmt3 mutant plants (Figure 7F). But there are a few concerns:

(a). The conclusion that RdDM/Pol IV is recruited to non-CG methylation regardless of H3K9me is mostly based on correlations. An alternative hypothesis favoring the dependency of H3K9me for RdDM could be formed based on the data in this manuscript. H3K9me is depleted at the DRM TEs and enriched in the CMT TEs in the h1 plants (Figure 1C). Do these changes in the H3K9me levels positively correlate with the mCHH changes at DRM and CMT TEs in the h1 plants? If so, does it suggest that H3K9me contributes to RdDM recruitment consistent with how sRNA accumulation positively correlates with H3K9me in the h1 plants (Figures 3A and B)?

(b). Most evidence supporting the conclusion that RdDM is recruited to non-CG methylation comes from analyses in h1 mutations. It remains largely unclear whether non-CG methylation explains recruitment at the DRM regions in the wt plants.

3. The authors conclude that SHH1 is dispensable for RdDM/Pol IV recruitment at CMT TEs in h1 plants based on the observation that mCHH levels at CMT TEs are similar in h1cmt2 and h1cmt2shh1 plants (Figure 3C). But in contrast to the largely invariable mCHH levels at the CMT TEs, regardless of the presence of SHH1 in the h1cmt2 background, sRNA accumulation at CMT TEs is reduced in shh1 or clsy1/2 mutants (see the sRNA levels in shh1 or clsy1/2 compared to wt in Figure 3C). This argues that the levels of sRNAs and mCHH do not quantitatively reflect each other at the CMT TEs. Could it be that the Pol IV branch and Pol V branch of the RdDM pathway are regulated differently at the CMT TEs versus the DRM TEs? It would be useful to compare the levels of mCHH and sRNAs in h1 nrpd1 and h1 nrpe1 mutants.

4. Another conclusion made in the manuscript is that sRNAs accumulating at CMT TEs in the h1 plants are mediated by CLSY3/4 and Pol IV complexes. It is shown in the manuscript that sRNA accumulation is elevated at CMT TEs in h1 plants (Figure 1D) and decreased in clsy3 clsy4 mutants (Figure 3C). sRNAs-seq analysis for a h1clsy3clsy4 mutant would be informative to test the need for CLSY3/4 for the over-accumulation of sRNAs in h1 plants.

5. The authors propose a model in which mCHG/mCHH marks promote and amplify Pol IV RdDM activity. While the data gathered do show that sRNA biogenesis correlates with mCHG/mCHH better than with H3K9me or mCG, an alternative hypothesis is that one single H3K9me or a small patch of mCG could initiate RdDM, which then amplifies sRNA biogenesis and cytosine methylation via a positive feedback loop. Thus, the issue is whether mCHG/mCHH correlation is a cause or effect of RdDM activity.

6. In double mutants involving the CG methyltransferase met1, one wonders how these mutants were generated and whether transgenerational epigenetic effects were considered. Blevins et al., showed in 2014 that MET1 and HDA6 are required for some loci to undergo RdDM and that in met1 or hda6 mutants, what they called "silent locus identity" is lost, presumably due to the loss of maintenance methylation. Once lost, silent locus identity is not regained by restoring MET1 or HDA6 activity. So, in experiments comparing met1 mutants to h1 met1 double mutants and looking for suppression of the met1 phenotype, it is important to consider the fact that suppression may not be possible, because silent locus identity was lost in the met1 mutant and can't be regained, regardless of secondary mutations (such as h1) combined with met1. A needed control is to see if the met1 phenotype can be complemented by restoring MET1 activity. If so, one can then have confidence in experiments with second mutations to see if they can counter suppress the met1 phenotype

Secondary issues:

1. One of the fundamental assumptions in the manuscript is that sRNA levels can be used as a proxy to measure RdDM activity in the designated CMT2 TEs in h1 mutants. While this is logical, it would be ideal if there were another stream of evidence supporting the expansion of RdDM to these areas, such as ChIP of a RdDM component (such as Pol IV) and see if Pol IV occupancy correlates with sRNA changes at CMT and DRM TEs.

2.The CHHs at DRM TEs appears to be hypermethylated in cmt2 mutants (mCHH panel in Figure 1—figure supplement 1A and mCHH panel in Figure 3C). Are there explanations for this?

3. It appears that sRNA profiles in the wt plants shown in Figure 1D and Figure 3C are very different. In Figure 1D, the sRNAs are clearly less enriched at the CMT regions than at the DRM regions in the wt plants. However, this difference is not seen in Figure 3C.

4. In Figure 5, it would be useful to categorize the loci into CMT and DRM regions as in Figures 1 and 3. It is important to know how H3K9me changes at the CMT TEs and DRM TEs with the mutations of h1 and ddm1.

5. Please explain how the well-positioned nucleosomes are defined in Figure 2. Why are some MNase-insensitive peaks defined as well-positioned nucleosomes and others are not (see the bottom track of Figure 2C)?

6. It is shown that the TE-edges and TE-bodies of heterochromatic TEs are differently targeted by RdDM and CMT2/3 in a former paper from the Zilberman lab (Zemach et al., 2013). Thus, CHH methylation at a long TE can be mediated by both RdDM and CMT2/3. To minimize the overlap of two pathways, the current manuscript categorized TEs based on their dependency on either DRM1/2 (mCHH > 0.05 in wt and mCHH < 0.02 in drm1drm2) or CMT2/3 (mCHH > 0.05 in wt and mCHH < 0.02 in cmt2cmt3). Alternatively, the heterochromatic TEs can be divided into TE edges and TE bodies as described in Zemach et al., 2013. Are the TE edges also hypomethylated (like those TEs defined as DRM TEs in the current manuscript) in h1 plants?

7. Singh et al., 2019 is appropriate to cite at the end of the following sentence in the introduction: "RdDM loci are transcribed by a methylation-tolerant RNA polymerase II derivative (Pol IV) that couples co-transcriptionally with RNA-dependent RNA polymerase 2 (RDR2) to make double stranded RNA, which is processed into 23/24-nt fragments by Dicer-like 3 (DCL3)." Several points in the sentence derive from that paper, not the review articles that are cited.

*Reviewer #2:*

RNAi-based DNA methylation, RdDM, is targeted to euchromatic transposable elements, but this targeting has been thought to be mediated by binding of an RdDM component, SHH1, to H3K9me, an epigenetic mark of heterochromatin. Here the authors show that the exclusion of RdDM from heterochromatic regions depends on histone H1. More unexpectedly, they show that targeting of RdDM to heterochromatic regions in the absence of H1 does not depend on H3K9me, or SHH1. Instead, the results suggest that RdDM is targeted to regions with non-CG methylation.

I enjoyed reading the manuscript very much. The conclusions are important and unexpected. Overall, the experiments are well designed and results are convincing. Below are my suggestions to strengthen the manuscript.

1) Based on the association of siRNA and non-CG methylation in h1c2c3, the authors discussed that RdDM is targeted to regions with non-CG methylation (Figure 7, 8). I assume that non-CG methylation remaining in h1c2c3 mutant is catalyzed by DRM, and wonder if this DRM-mediated non-CG methylation in h1c2c3 can be consequence, rather than trigger, of RdDM. This possibility could be discussed, or excluded. In addition, according to the results of Stroud et al., 2014 (Figure 5), ddcc mutation results in drastic loss of sRNA in DRM targets, but the effects are smaller in some of CMT targets. Does that suggest an additional (perhaps minor) pathway to target RdDM in the background of complete loss of non-CG methylation in the ddcc mutant? That could be discussed.

2) If the authors have any idea about RdDM component(s) that recognize non-CG methylation and recruit RdDM machinery to heterochromatic regions, that might be discussed.

3) clsy1/2 and clsy3/4 affect RdDM in targets of DRM and CMT, respectively (Figure 3C). Do their target spectra change in the h1 background? I wonder if the differential effects of CLSYs are also defined by H1, or defined directly by GC content etc.

*Reviewer #3:*

Two general types of TEs exist in Arabidopsis: heterochromatic, GC-rich autonomous Tes (CMT Tes), and euchromatic, AT-rich, short non-autonomous Tes that tend to be proximal to genes (DRM Tes). CMT2 and CMT3 catalyze non-CG methylation at heterochromatic Tes, and are recruited by H3K9me2, which recruits SUVH H3K9 methyltransferases forming a feedback loop. RdDM catalyzes non-CG methylation on euchromatic Tes. Pol V is recruited to methylated DNA while Pol IV has been proposed to be recruited by H3K9me. Moreover, SHH binds H3K9me/H3K9me2 and is required for sRNA production at many loci. But if H3K9me recruits Pol IV, then how is Pol IV excluded from heterochromatic Tes that also have high H3K9me? And why would RdDM depend on H3K9me when RdDM targets themselves are depleted of H3K9me? In this manuscript, Choi et al., address this long-standing issue of how CMT TEs and DRM TEs are independently targeted for methylation.

Consistent with previous reports, sRNAs are redistributed from DRM TEs to CMT TEs in h1 mutants, and similar trends were observed for H3K9me2, mCHG and mCHH. Importantly, this DRM-to-CMT TE shift in h1 plants was also observed in h1 cmt2, which indicates that CHH hypermethylation is caused by hyper RdDM. Choi et al., also showed that H1 prevents sRNA production from TEs with very high H1 and H3K9me. SHH1 has been implicated in H3K9me binding and Pol IV recruitment, but CMT TEs remained hypermethylated in h1 cmt2 shh1 plants suggesting that Pol IV is recruited to CMT TEs independently of SHH1. Together with their h1 ddm1 results, this suggest that H3K9me is not required for Pol IV recruitment to CMT TEs. Further, their analyses also suggest that SUVH4/5/6 are recruited to mCG in the absence of non-CG methylation. Therefore, their results support that non-CG methylation helps recruit Pol IV.

H1 is the major factor excluding Pol IV from heterochromatin, which is consistent with previous reports. Contrary to the model that Pol IV is recruited by H3K9me or mCG, Choi et al., were able to show that non-CG methylation promotes Pol IV activity by analyzing multiple h1c2c3, suvh4/5/6 and polv mutants. This work demonstrates that RdDM is in a feedback loop with its product (i.e. non-CG methylation), and thus is not connected to CMT-SUVH pathway. This is consistent with RdDM being dedicated to maintaining methylation at shorter sequences where stable histone-based epigenetic inheritance is not possible. The authors end with a discussion about how the preference of H1 for GC-rich sequences may have evolved in plants so that H1 prefers GC-rich TEs that have higher coding capacity and allows RdDM to silence short AT-rich TEs.

Overall, I found this work well-designed, well-executed and clearly presented. Although it can be difficult to interpret direct and indirect effects of the higher-order mutants they analyzed, I agree with their main conclusion that "H1 enforces the separation of euchromatic and heterochromatic DNA methylation pathways by excluding the small RNA-generating branch of RdDM from non-CG methylated heterochromatin".

The manuscript may be strengthened by explicitly mentioning the limitations of their genome-wide analyses in regard to direct or indirect effects of the mutants they are analyzing. Additionally or alternatively, the authors could describe future experiments to test whether non-CG methylation is sufficient for sRNA production (e.g. analysis of sRNAs in mutants with increased mCHG).

---

## [Author Response]

Reviewer #1:This paper examines RNA-directed DNA methylation (RdDM) at transposable elements (TEs) in wild-type plants versus plants with defective mutations in genes encoding linker histone H1, the CG DNA methyltransferase MET1, the CHG and CHH DNA methyltransferases CMT2 and CMT3, the de novo DNA methyltransferases DRM1 and DRM2, the chromatin remodeling enzyme DDM1, the putative chromatin remodeling proteins CLSY1, CLSY2, CLSY3 and CLSY4, the histone H3 lysine 9 methyl group (H3K9me) binding protein SHH1, the Histone H3 lysine 9 methyltransferases SUVH4,5 and 6 and the DNA-dependent RNA polymerases Pol IV and Pol V. This is an extensive list of mutants, which are examined in a number of new combinations. Mutations combined with the loss of histone H1 yielded several new insights. Without histone H1, TEs whose methylation primarily depends on CMT2/3 (CMT TEs) were found to gain methylation whereas TEs that are mainly methylated by DRM1/2 became hypomethylated. H3K9me levels are typically high at DNA loci with high levels of cytosine methylation, and SHH1 is a protein that can bind H3K9me and is implicated in recruitment of Pol IV to initiate the small RNA (sRNA) part of the RdDM pathway. Contrary to current thinking, the authors argue that H3K9 methylation binding by SHH1 plays a small role in recruiting RdDM machinery to CMT TEs in h1 mutants, as loss of SHH1 did not diminish an increase sRNAs and CHH hypermethylation at CMT TEs. The authors present evidence that sRNA biogenesis at CMT TEs in h1 mutants requires CLSY3/4, as a large proportion of hypermethylated TEs in h1cmt2 and h1cmt2shh1 mutants overlaps with the CLSY3/4-dependent sRNAs. To test if CLSY3/4 recruitment is dependent on CG methylation, as previously proposed, a h1met1 line was used. The authors found that the loss of CG methylation and histone H1 in h1met1 mutants resulted in an increase in methylation at CHH motifs instead of the expected decrease, suggesting that CG methylation is not the chromatin mark key to CLSY3/4-dependent RdDM detected at heterochromatic TEs in h1 mutants. The authors propose instead that methylated mCHG/mCHH motifs are the primary marks for recruiting the RdDM machinery, with H3K9me-dependent mechanisms (e.g. via SHH1 and CLSY1/2) playing a lesser role.Many of the claims are supported by the data presented and are convincing; however, some claims, including the conclusion, need more data or further clarification.Primary issues to be addressed:1. The authors hypothesize that histone H1 prohibits RdDM from occurring at CMT2-dependent heterochromatic regions, which have high levels of H1 association. This is supported by the observation that plants with mutations in H1 genes (h1) exhibit elevated mCHH levels at these loci. In contrast, histone H1 is depleted in DRM-dependent regions (Figure 1—figure supplement 1), which is hypothesized to allow RdDM to occur. One might expect mCHH levels to increase at DRM-dependent regions in h1 mutations, but CHH motifs at the DRM-dependent regions become hypomethylated. The authors speculate that dilution of RdDM machinery by expanding to CMT regions may contribute to the reduced mCHH at the DRM TEs. However, the 'dilution' hypothesis is highly speculative and not supported by direct evidence. The "dilution" idea brought to mind a 2012 study by Wierzbicki et al., that found that total levels of CHH methylation remained fairly constant when wild-type plants, pol IV mutant plants or pol V mutant plants were compared, but with CHH methylation increasing in pericentromeric regions and decreasing in chromosome arms in the pol IV/V mutants. An increase in CHH methylation in pericentric regions due to increased RdDM seems unlikely as an explanation for the 2012 results, as RdDM is expected to be eliminated in pol IV and pol V mutants. Instead, the pericentromeric methylation is presumably attributable to CMT2 activity based on the current and former studies of the Zilberman lab. Collectively, these old and new results suggests that CMT2 and RdDM-dependent methylation may involve competition for a common factor present in limiting concentration. As an example, one candidate might be the methyl donor S-adenosyl methionine (AdoMet). If there is competition for a limiting substrate, knocking out either the CMT or DRM-dependent methylation pathway may simply make more AdoMet available to the other pathway, resulting in higher levels of methylation. The authors should consider such alternative hypothesis, as opposed to the spreading of the machinery for one pathway into new territories. Conducting ChIP experiments would also allow tests for the authors’ spreading hypothesis.

We show that sRNA levels and mCHH are increased in *h1* heterochromatin, and that this does not require CMT2. Therefore, our results cannot be explained by increased CMT activity in the absence of H1, although increased CMT activity is almost certainly a consequence of H1 loss, as evidenced by major mCHG gains in *h1* heterochromatin presumably catalyzed by CMT3. The only plausible explanation for our results is increased RdDM activity in heterochromatin. This is explained in the Results section:

“CMT TE mCHH increases to the same relative extent in *h1* plants devoid of CMT2 (*h1c2*; Figure 1F and Figure 1—figure supplement 1H), indicating that mCHH hypermethylation at CMT TEs in *h1* mutants is caused by RdDM. These results indicate that RdDM relocates into heterochromatin in the absence of H1 and are consistent with recently published work (Bourguet et al., 2021; Papareddy et al., 2020).”

Furthermore, we show that mutation of CMT2 and CMT3 in *h1c2c3* mutants restores RdDM activity to something like a *wt* state. If the CMT and RdDM pathways are simply competing for a resource, loss of CMT2/3 should activate RdDM (as proposed by the Reviewer), not abrogate RdDM in heterochromatin and return it to its normal euchromatic targets (as we observe). Instead, our data are most compatible with a model in which mCHH/CHG attracts and H1 inhibits Pol IV activity. We now explicitly emphasize the rescue of RdDM patterns in *h1c2c3* mutants in Figure 8G-H and Figure 8—figure supplement 1E-G, and discuss the significance of these results:

“The key observations are that loss of CMT2/3 in *h1c2c3* plants (and the associated loss of mCHG/mCHH) largely abrogates the relocation of Pol IV activity into heterochromatin (Figure 8G-H and Figure 8—figure supplement 1E-G), and the remaining heterochromatic sRNA biogenesis is not associated with H3K9me or mCG (Figure 8D-F). These results do not support the hypothesis that Pol IV is recruited by H3K9me, and offer non-CG methylation as the most likely alternative. Our data suggest the hypothesis that without H1, non-CG methylation catalyzed by CMT2/3 pulls Pol IV into heterochromatin, and loss of CMT2/3 allows Pol IV to return to its mostly euchromatic *wt* targets.”

The increased heterochromatic mCHH in RdDM mutants is most likely caused by increased SUVH4/5/6 activity in heterochromatin caused by loss of euchromatic mCHH/CHG. As AdoMet is a methyl donor involved in myriad cytoplasmic and nuclear methylation reactions, we think that loss of RdDM is unlikely to significantly alter its availability to other enzymatic processes.

2. One major conclusion in the manuscript is that RdDM/Pol IV is recruited to non-CG methylated loci regardless of H3K9 methylation. Multiple observations support this conclusion: (a). CMT Tes are similarly hypermethylated in h1cmt2shh1 and h1cmt2 plants (Figure 3C), suggesting that the CHH methylation at the CMT regions are mediated independently of SHH1, which is a H3K9me binding protein thought to mediate Pol IV recruitment; (b). sRNAs hyper-accumulate in a h1ddm1 mutant despite a large decrease in H3K9me in this background (Figure 5); (c). sRNA accumulation better correlates with non-CG methylation than with H3K9me in the h1cmt2cmt3 mutant plants (Figure 7F). But there are a few concerns:a). The conclusion that RdDM/Pol IV is recruited to non-CG methylation regardless of H3K9me is mostly based on correlations. An alternative hypothesis favoring the dependency of H3K9me for RdDM could be formed based on the data in this manuscript. H3K9me is depleted at the DRM Tes and enriched in the CMT Tes in the h1 plants (Figure 1C). Do these changes in the H3K9me levels positively correlate with the mCHH changes at DRM and CMT Tes in the h1 plants? If so, does it suggest that H3K9me contributes to RdDM recruitment consistent with how sRNA accumulation positively correlates with H3K9me in the h1 plants (Figures 3A and B)?

mCHH/CHG and H3K9me levels almost always correlate, which is why they are so difficult to disentangle. For most of the paper, our data (including the data in Figures 1 and 3) are equally consistent with H3K9me or mCHH/CHG mediating Pol IV recruitment. Only analyses that decouple H3K9me2 and mCHH/CHG (Figures 8 and 9) can distinguish whether one or the other mediates Pol IV activity. These analyses show that H3K9me alone is not a good correlate of Pol IV activity.

b). Most evidence supporting the conclusion that RdDM is recruited to non-CG methylation comes from analyses in h1 mutations. It remains largely unclear whether non-CG methylation explains recruitment at the DRM regions in the wt plants.

We believe that some of the most convincing results in the paper linking RdDM recruitment to non-CG methylation are in the Figure 9, most of which (panels B-H) shows analyses of *suvh4/5/6* and *pol v* mutants. These analyses demonstrate that *h1* mutations are not required to identify a link between Pol IV activity and non-CG methylation (as opposed to H3K9me).

3. The authors conclude that SHH1 is dispensable for RdDM/Pol IV recruitment at CMT Tes in h1 plants based on the observation that mCHH levels at CMT Tes are similar in h1cmt2 and h1cmt2shh1 plants (Figure 3C). But in contrast to the largely invariable mCHH levels at the CMT Tes, regardless of the presence of SHH1 in the h1cmt2 background, sRNA accumulation at CMT Tes is reduced in shh1 or clsy1/2 mutants (see the sRNA levels in shh1 or clsy1/2 compared to wt in Figure 3C). This argues that the levels of sRNAs and mCHH do not quantitatively reflect each other at the CMT Tes. Could it be that the Pol IV branch and Pol V branch of the RdDM pathway are regulated differently at the CMT Tes versus the DRM Tes? It would be useful to compare the levels of mCHH and sRNAs in h1 nrpd1 and h1 nrpe1 mutants.

Levels of sRNA and mCHH/CHG do quantitatively reflect each other at CMT Tes when the relevant CMT enzymes are inactivated (see new Figure 8). This is a major reason why we chose to analyze *h1cmt2shh1* mutants as opposed to *h1shh1* mutants. The relationship is more complicated when CMT enzymes that can catalyze mCHH/CHG are present (compare correlations between sRNA and mCHH/CHG in Figure 8B and Figure 8D). Hence, sRNA levels at CMT Tes can decrease in *shh1* or *clsy1/2* mutants, whereas mCHH in these mutants increases at CMT Tes (former Figure 3C; new Figure 4A).

4. Another conclusion made in the manuscript is that sRNAs accumulating at CMT Tes in the h1 plants are mediated by CLSY3/4 and Pol IV complexes. It is shown in the manuscript that sRNA accumulation is elevated at CMT Tes in h1 plants (Figure 1D) and decreased in clsy3 clsy4 mutants (Figure 3C). sRNAs-seq analysis for a h1clsy3clsy4 mutant would be informative to test the need for CLSY3/4 for the over-accumulation of sRNAs in h1 plants.

Our data show that sRNA accumulation at CMT Tes does not require *shh1*. Given the published strong link between SHH1 and CLSY1/2, and the requirement of CLSYs for sRNA biogenesis, we concluded that CLSY3/4 mediate the relocation of Pol IV to CMT Tes in *h1* plants. However, it is formally possible that this process is mediated by CLSY1/2 independently of SHH1 or does not require any CLSY activity. We now mention these possibilities in the Results section:

“However, our results do not rule out the possibility that some of the RdDM expansion in *h1* plants is mediated by CLSY1/2 or is independent of CLSY activity.”

5. The authors propose a model in which mCHG/mCHH marks promote and amplify Pol IV RdDM activity. While the data gathered do show that sRNA biogenesis correlates with mCHG/mCHH better than with H3K9me or mCG, an alternative hypothesis is that one single H3K9me or a small patch of mCG could initiate RdDM, which then amplifies sRNA biogenesis and cytosine methylation via a positive feedback loop. Thus, the issue is whether mCHG/mCHH correlation is a cause or effect of RdDM activity.

With a functional Pol V pathway, mCHH/CHG will of course be a product of RdDM activity. The core mystery our study set out to explore is why this doesn’t produce a good correlation between sRNA and mCHH/CHG in wild-type plants. In plants without CMT2/3, some correlation is effectively guaranteed, because all mCHH/CHG is a product of RdDM. What matters with respect to mCG is that nearly complete lack of mCG in *met1* does not perturb sRNA biogenesis so long as mCHH/mCHG is present, ruling out mCG as a significant determinant of Pol IV activity. With respect to H3K9me, what matters is that sRNA biogenesis does not correlate with this modification in *h1c2c3* plants in which the link between mCHH/CHG and H3K9me is broken. We now explain this explicitly in the results:

“It is important to note that in plants lacking CMT2/3, all mCHH should be catalyzed by RdDM, and a correlation between sRNA (product of the Pol IV pathway) and mCHH (product of the Pol V pathway) is therefore expected regardless of how Pol IV is recruited. The key observations are that loss of CMT2/3 in *h1c2c3* plants (and the associated loss of mCHG/mCHH) largely abrogates the relocation of Pol IV activity into heterochromatin (Figure 8G-H and Figure 8—figure supplement 1E-G), and the remaining heterochromatic sRNA biogenesis is not associated with H3K9me or mCG (Figure 8D-F). These results do not support the hypothesis that Pol IV is recruited by H3K9me, and offer non-CG methylation as the most likely alternative. Our data suggest the hypothesis that without H1, non-CG methylation catalyzed by CMT2/3 pulls Pol IV into heterochromatin, and loss of CMT2/3 allows Pol IV to return to its mostly euchromatic *wt* targets.”

Furthermore, analysis of *pol v* mutants, in which non-CG methylation is not caused by RdDM, shows that neither mCG nor H3K9me can explain sRNA biogenesis. We now make this point more explicitly in the results:

“Finally, we assayed CLSY1/2 clusters with low *wt* H3K9me2 but high *wt* sRNA and mCHH (LH CLSY1/2 clusters) in *polv* mutants to determine whether non-CG methylation is required to maintain sRNA expression. RNA Pol V is not directly involved in sRNA production, but is an essential RdDM component required for DNA methylation because it recruits DRM2 (Erdmann and Picard, 2020; Matzke and Mosher, 2014; Raju et al., 2019; Wendte and Pikaard, 2017). Therefore, *polv* mutants allow us to differentiate non-CG methylation as a cause vs. a consequence of Pol IV activity. 90% of the 662 LH CLSY1/2 clusters lose mCHH in *polv* plants (mCHH<0.05, Figure 9E), and the overall non-CG methylation of LH CLSY1/2 clusters is greatly reduced without Pol V (Figure 9F). In *suv4/5/6* mutants, LH CLSY1/2 clusters maintain sRNA expression, whereas sRNA expression in *polv* mutants is greatly reduced (Figure 9G). Furthermore, mCG at LH CLSY1/2 clusters is higher in *polv* than in *suvh4/5/6* plants (Figure 9H). Therefore, sRNA biogenesis is not sensitive to the loss of either H3K9me2 or mCG and specifically requires non-CG methylation.”

6. In double mutants involving the CG methyltransferase met1, one wonders how these mutants were generated and whether transgenerational epigenetic effects were considered. Blevins et al., showed in 2014 that MET1 and HDA6 are required for some loci to undergo RdDM and that in met1 or hda6 mutants, what they called "silent locus identity" is lost, presumably due to the loss of maintenance methylation. Once lost, silent locus identity is not regained by restoring MET1 or HDA6 activity. So, in experiments comparing met1 mutants to h1 met1 double mutants and looking for suppression of the met1 phenotype, it is important to consider the fact that suppression may not be possible, because silent locus identity was lost in the met1 mutant and can't be regained, regardless of secondary mutations (such as h1) combined with met1. A needed control is to see if the met1 phenotype can be complemented by restoring MET1 activity. If so, one can then have confidence in experiments with second mutations to see if they can counter suppress the met1 phenotype

We are, of course, aware of the epigenetic properties of mCG. When we work with mutants that substantially perturb mCG (*met1* or *ddm1*), we always keep them heterozygous until all other mutations are fixed. However, we do not see how any form of *h1met1* mutant generation would make a difference to our conclusions. We use *h1met1* to show that TEs that have lost mCG and mCHH/CHG (lost “silent locus identity”) have unaltered (low) levels of sRNA compared to *met1*, whereas TEs that lost mCG but retained mCHH/CHG gain sRNA compared to *met1*. This leads us to conclude that mCG is dispensable for sRNA accumulation caused by loss of H1. Nothing about this conclusion would be altered by results from a MET1 complementation experiment.

Secondary issues:1. One of the fundamental assumptions in the manuscript is that sRNA levels can be used as a proxy to measure RdDM activity in the designated CMT2 TEs in h1 mutants. While this is logical, it would be ideal if there were another stream of evidence supporting the expansion of RdDM to these areas, such as ChIP of a RdDM component (such as Pol IV) and see if Pol IV occupancy correlates with sRNA changes at CMT and DRM TEs.

We feel that sRNA levels and mCHH/CHG are the most relevant measures of RdDM activity, being the outputs of the Pol IV and Pol V pathways, respectively. Although we agree that ChIP of an RdDM component might be informative, we do not have the required lines or antibodies. Therefore, as such experiments would take substantial time and are very unlikely to alter any of the main conclusions of an already long and complex paper, we prefer to forgo additional ChIP-seq experiments.

2.The CHHs at DRM TEs appears to be hypermethylated in cmt2 mutants (mCHH panel in Figure 1—figure supplement 1A and mCHH panel in Figure 3C). Are there explanations for this?

The most likely explanation is that loss of heterochromatic mCHH in *cmt2* mutants liberates RdDM machinery for greater activity at DRM TEs.

3. It appears that sRNA profiles in the wt plants shown in Figure 1D and Figure 3C are very different. In Figure 1D, the sRNAs are clearly less enriched at the CMT regions than at the DRM regions in the wt plants. However, this difference is not seen in Figure 3C.

The profiles are indeed different. Our sRNA data (Figure 1D) come from leaves, which have lower enrichment of sRNA at heterochromatic CLSY3/4 loci. The published data in former Figure 3C (new Figure 4A) are from flowers, which have much more heterochromatic CLSY3/4 sRNA.

Throughout the paper, we have been careful to pair mutant data with the appropriate wild-type control. We now explain this in the Results section and include a new supplementary figure to demonstrate why the choice of control is crucial:

“Also, please note that the *wt* sRNA patterns in Figures 1D and 4A are distinct because the former is from leaves and the latter from inflorescences. Leaf sRNA levels are lower at CMT TEs and CLSY3/4 clusters compared to flowers (Figure 4—figure supplement 2), presumably due to higher expression of CLSY3/4 in reproductive tissues (Long et al., 2021; Zhou et al., 2021).”

4. In Figure 5, it would be useful to categorize the loci into CMT and DRM regions as in Figures 1 and 3. It is important to know how H3K9me changes at the CMT TEs and DRM TEs with the mutations of h1 and ddm1.

We altered former Figure 5 (new Figure 6) accordingly.

5. Please explain how the well-positioned nucleosomes are defined in Figure 2. Why are some MNase-insensitive peaks defined as well-positioned nucleosomes and others are not (see the bottom track of Figure 2C)?

We use nucleosome positioning data and definitions from a previous manuscript (Lyons and Zilberman, *eLife* 2017). This paper describes in detail how well-positioned nucleosomes are defined. In short, correspondence between biological replicates is they key measure of nucleosome positioning. Some MNase-insensitive peaks may not be defined as well-positioned if the correspondence between biological replicates is low. We added the following sentence to the legend of Figure 2: “Nucleosome positioning data and designations are from (Lyons and Zilberman, 2017).”

6. It is shown that the TE-edges and TE-bodies of heterochromatic TEs are differently targeted by RdDM and CMT2/3 in a former paper from the Zilberman lab (Zemach et al., 2013). Thus, CHH methylation at a long TE can be mediated by both RdDM and CMT2/3. To minimize the overlap of two pathways, the current manuscript categorized TEs based on their dependency on either DRM1/2 (mCHH > 0.05 in wt and mCHH < 0.02 in drm1drm2) or CMT2/3 (mCHH > 0.05 in wt and mCHH < 0.02 in cmt2cmt3). Alternatively, the heterochromatic TEs can be divided into TE edges and TE bodies as described in Zemach et al., 2013. Are the TE edges also hypomethylated (like those TEs defined as DRM TEs in the current manuscript) in h1 plants?

TE edges indeed behave like DRM TEs, which can be seen in the genome browser snapshots in Figures 1E and 8H (former Figure 7G). We now include a more extensive analysis of TE edges in Figure 1—figure supplement 1F-G, Figure 6D, Figure 8G, and Figure 8—figure supplement 1E.

7. Singh et al., 2019 is appropriate to cite at the end of the following sentence in the introduction: "RdDM loci are transcribed by a methylation-tolerant RNA polymerase II derivative (Pol IV) that couples co-transcriptionally with RNA-dependent RNA polymerase 2 (RDR2) to make double stranded RNA, which is processed into 23/24-nt fragments by Dicer-like 3 (DCL3)." Several points in the sentence derive from that paper, not the review articles that are cited.

Singh et al., 2019 is now cited as suggested by the reviewer.

Reviewer #2:RNAi-based DNA methylation, RdDM, is targeted to euchromatic transposable elements, but this targeting has been thought to be mediated by binding of an RdDM component, SHH1, to H3K9me, an epigenetic mark of heterochromatin. Here the authors show that the exclusion of RdDM from heterochromatic regions depends on histone H1. More unexpectedly, they show that targeting of RdDM to heterochromatic regions in the absence of H1 does not depend on H3K9me, or SHH1. Instead, the results suggest that RdDM is targeted to regions with non-CG methylation.I enjoyed reading the manuscript very much. The conclusions are important and unexpected. Overall, the experiments are well designed and results are convincing. Below are my suggestions to strengthen the manuscript.1) Based on the association of siRNA and non-CG methylation in h1c2c3, the authors discussed that RdDM is targeted to regions with non-CG methylation (Figure 7, 8). I assume that non-CG methylation remaining in h1c2c3 mutant is catalyzed by DRM, and wonder if this DRM-mediated non-CG methylation in h1c2c3 can be consequence, rather than trigger, of RdDM. This possibility could be discussed, or excluded. In addition, according to the results of Stroud et al., 2014 (Figure 5), ddcc mutation results in drastic loss of sRNA in DRM targets, but the effects are smaller in some of CMT targets. Does that suggest an additional (perhaps minor) pathway to target RdDM in the background of complete loss of non-CG methylation in the ddcc mutant? That could be discussed.

Please see our response to Reviewer 1’s main point #5 regarding the issue of non-CG methylation being both a cause and consequence of RdDM. This is an important point, and we hope we have now clarified it in the paper.

We thank the reviewer for pointing out the published *ddcc* results. We did not mean to argue that non-CG methylation is absolutely required for sRNA biogenesis, as the published *ddcc* data make clear. We now state this explicitly in the Discussion section:

“Our data linking 24-nt biogenesis with non-CG methylation do not mean that such methylation is absolutely required for Pol IV recruitment. Indeed, there is residual 24-nt biogenesis in *ddcc* mutants that lack non-CG methylation (Stroud et al., 2014). One possibility is that the factor or factors recruiting Pol IV to non-CG methylation have weak affinity for mCG, which could recruit Pol IV in the absence of non-CG methylation, analogous to our proposed mode of SUVH4/5/6 recruitment in plants lacking CMT2/3. Other chromatin features may also recruit or facilitate Pol IV activity. However, our results indicate that non-CG methylation is the major Pol IV recruiting genomic feature under normal conditions.”

2) If the authors have any idea about RdDM component(s) that recognize non-CG methylation and recruit RdDM machinery to heterochromatic regions, that might be discussed.

We agree that identifying such component(s) would be very important, but unfortunately we do not have anything useful to contribute on this point.

3) clsy1/2 and clsy3/4 affect RdDM in targets of DRM and CMT, respectively (Figure 3C). Do their target spectra change in the h1 background? I wonder if the differential effects of CLSYs are also defined by H1, or defined directly by GC content etc.

Our data do not argue that H1 alters the balance of preference between CLSY1/2 and CLSY3/4 for various loci. However, substantial additional data would be required to clearly address this question. Specifically, we would need data at least from *h1clsy1/2* and *h1clsy3/4* lines. We are reluctant to speculate about this matter in the paper without these data.

Reviewer #3:[…]The manuscript may be strengthened by explicitly mentioning the limitations of their genome-wide analyses in regard to direct or indirect effects of the mutants they are analyzing. Additionally or alternatively, the authors could describe future experiments to test whether non-CG methylation is sufficient for sRNA production (e.g. analysis of sRNAs in mutants with increased mCHG).

We now include new analysis of *ibm1* mutant data (new Figure 3C-D) that shows gain of sRNA production associated with gain of mCHG/CHH and H3K9me. This result does not distinguish between non-CG methylation and H3K9me but does show that gain of one or both is sufficient to activate sRNA production.

We now include a statement in the Discussion section about the limitations of our conclusions linking non-CG methylation with sRNA biogenesis, as we describe above in response to point #1 of Reviewer 2.